# Improved modeling of human vision by incorporating robustness to blur in convolutional neural networks

Hojin Jang [1,2,3] ✉ & Frank Tong [1] ✉

Whenever a visual scene is cast onto the retina, much of it will appear degraded due to poor resolution in the periphery; moreover, optical defocus can cause blur in central vision. However, the pervasiveness of blurry or degraded input is typically overlooked in the training of convolutional neural networks (CNNs). We hypothesized that the absence of blurry training inputs may cause CNNs to rely excessively on high spatial frequency information for object recognition, thereby causing systematic deviations from biological vision. We evaluated this hypothesis by comparing standard CNNs with CNNs trained on a combination of clear and blurry images. We show that blur-trained CNNs outperform standard CNNs at predicting neural responses to objects across a variety of viewing conditions. Moreover, blur-trained CNNs acquire increased sensitivity to shape information and greater robustness to multiple forms of visual noise, leading to improved correspondence with human perception. Our results provide multi-faceted neurocomputational evidence that blurry visual experiences may be critical for conferring robustness to biological visual systems.

A hallmark of human vision lies in its robustness to challenging or ambiguous viewing conditions. Consider the difficulties of navigating traffic in a snowstorm, detecting a pedestrian in the corner of one's eye, or identifying a distant building that is veiled in fog. In laboratory settings, researchers have characterized the robustness of human object recognition to visual noise, blur, and other forms of image degradation[1–6]. Our ability to recognize objects depends on the ventral visual pathway, which extends from early visual areas (V1-V4) to higher level object-sensitive areas in the occipitotemporal cortex[7–14]. Neuroimaging studies have investigated the functional selectivity and topographical organization of the ventral visual system, informing our understanding of the neural bases of object recognition under clear viewing conditions[9,15] as well as conditions of visual ambiguity[16–18].

To understand the neurocomputational bases of object recognition, researchers have sought to develop computational models that can effectively predict the visual system's responses to complex objects. Indeed, recent studies have found that deep convolutional neural networks (CNNs) trained on tasks of object recognition provide the best current models of the visual system, allowing for reliable prediction of visual cortical responses in humans[19–24] and neuronal responses in the macaque inferotemporal cortex[25–29]. While these initial findings are highly promising, a mounting concern is that CNNs tend to catastrophically fail where humans do not, especially when presented with noisy, blurry or otherwise degraded visual stimuli[5,6,30–32]. Such findings demonstrate that the computations and learned representations of these CNNs are not truly aligned with those of the human brain.

We considered the susceptibility of CNNs to visual blur to be of particular interest, as blur is pervasive in everyday human vision[33,34]. A common misperception is that our visual world is entirely clear, when much of what we see is either blurry or processed with low spatial resolution. The density of cone photoreceptors, bipolar

[1]Department of Psychology, Vanderbilt Vision Research Center, Vanderbilt University, Nashville, TN, USA. [2]Department of Brain and Cognitive Sciences, Massachusetts Institute of Technology, Cambridge, MA, USA. [3]Present address: Department of Brain and Cognitive Engineering, Korea University, Seoul, South Korea. ✉e-mail: hojin4671@korea.ac.kr; frank.tong@vanderbilt.edu

neurons, and ganglion cells decreases precipitously from the fovea to the periphery; thus, only stimuli that appear near the center of gaze can be processed with high spatial resolution[35]. To capitalize on the much higher resolution of the fovea, humans make multiple eye movements every second to bring objects of interest to the center of gaze[36,37]. However, eye movements that involve large changes in vergence will cause foveated objects to initially appear blurry, due to the sluggish nature of lens accommodation[38,39]. Moreover, even after a central object is accurately fixated and accommodated, a parafoveal object that appears at a different depth plane may also appear blurry due to defocus aberration[33,34]. Thus, low-resolution vision and blur are prominent features of everyday vision. By contrast, the image datasets commonly used to train CNNs predominantly consist of clear, well-focused images[40,41].

There is a general bias to consider blurry vision as suboptimal, problematic, and in need of correction. However, such assumptions may overlook the potential contributions of blur for real-world vision and object recognition. For example, humans can leverage blurry contextual information to support more accurate object recognition[17,42], and both face and object recognition remain quite robust to substantial levels of blur[3,6,43]. Recent work has further revealed that defocus blur provides an important cue for depth perception[44]. Neurophysiological studies have also found that shape-selective neurons in the visual cortex can be tuned to varying degrees of blur, with some neurons preferring blurry over clear depictions of 2D object shapes[45]. Thus, blur appears to be an important feature that is encoded by the visual system.

Here, we evaluated the hypothesis that the omission of blurry training inputs may cause CNNs to rely excessively on high spatial frequency information for object recognition, thereby causing systematic deviations from biological vision. To address this question, we compared the performance of standard CNNs trained exclusively on clear images with CNNs trained on a combination of clear and blurry images. By testing standard versus blur-trained CNNs on a diverse set of neural, visual, and behavioral benchmarks, we show that blur-trained CNN models significantly outperform standard CNNs at predicting neural responses to object images across a variety of viewing conditions, including those that were never used for training.

Unlike standard CNNs, blur-trained CNNs favor the processing of lower spatial frequency information, allowing for greater sensitivity to global object shape. Finally, contrary to the notion that CNNs are very poor at recognizing objects in novel or challenging viewing conditions, we show that CNNs trained on clear and Gaussian-blurred images exhibit greater robustness to multiple forms of blur, visual noise, as well as various types of image compression. From these findings, we conclude that instances of blurry vision are not fundamentally problematic for biological vision; instead blur may constitute a positive feature that can promote the development of more robust object recognition in both artificial and biological visual systems.

## Results

### Neural predictivity of standard and blur-trained CNNs

We compared the performance of 8 standard CNN models trained on clear images only with the same CNN architectures trained using either weak or strong levels of blur (see Fig. 1). For weak-blur CNN training, clear images ($\sigma = 0$) occurred with much greater frequency than blurry images to mimic the extent to which defocus blur would likely occur for future saccade targets in natural viewing tasks[34]. For the strong-blur CNNs, a Gaussian blur kernel of varying size ($\sigma = 0, 1, 2, 4$ or $8$ pixels) was applied with equal probability to the training images. This manipulation was informed by the fact that visual acuity systematically declines from the fovea to the periphery[35,46], such that varying degrees of spatial resolution are always present in one's visual experience. Further details regarding CNN model training can be found in the Methods.

We first sought to compare standard and blur-trained CNNs on their ability to account for functional magnetic resonance imaging (fMRI) responses obtained from the human visual cortex while observers viewed clear, low-pass or high-pass filtered object images by analyzing the data from a publicly available neuroimaging dataset[47]. To do so, we performed representational similarity analysis (RSA) on the response patterns to the set of object images that were found in individual human visual areas and in each layer of a CNN[15,19], and then computed the Pearson correlational similarity between the two RSA matrices of interest. Peak correlations were typically observed in an intermediate CNN layer (Supplementary Fig. 1) and this value was used

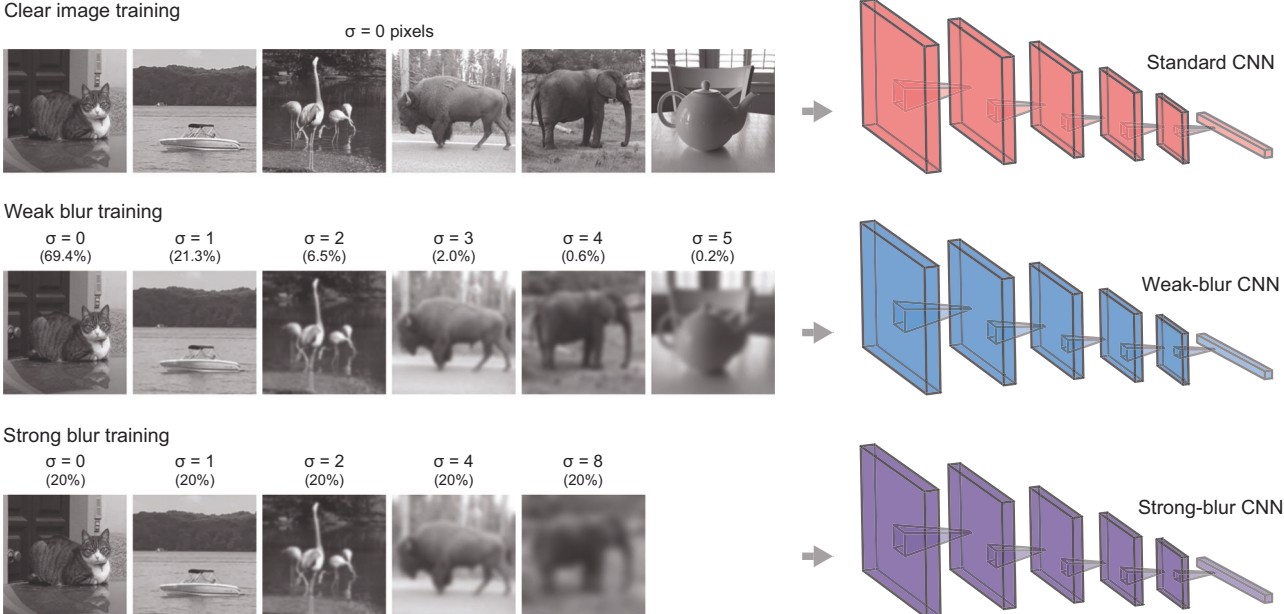

**Fig. 1 | Examples of images used for 3 different CNN training paradigms.** For weak-blur and strong-blur CNNs, object images were blurred by Gaussian kernels of varying width and presented with varying frequencies, as indicated. Images from the authors.

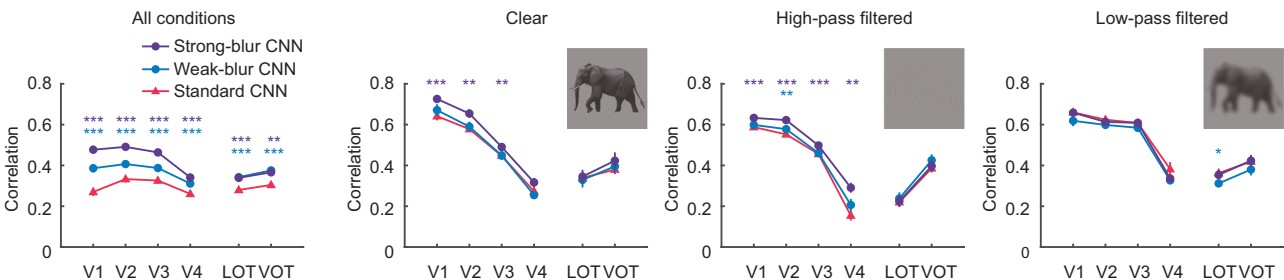

**Fig. 2 | Correlational similarity between CNN model responses and neural responses in different human visual areas to clear, high-pass filtered and low-pass filtered images.** A set of 8 different standard CNNs (red), weak-blur CNNs (blue) and strong-blur CNNs (purple) were evaluated using human fMRI data ($n = 10$) obtained from Xu and Vaziri-Pashkam (2021). Error bars represent ±1 standard error of the mean (SEM). Two-tailed paired t-tests were used to compare the model performance of blur-trained CNNs versus standard CNNs (*$p < 0.05$, **$p < 0.01$, and ***$p < 0.001$, uncorrected for multiple comparisons; the exact $p$ values and raw values are provided in the Source Data). Elephant images from the original publication with permission. LOT lateral occipitotemporal cortex, VOT ventral occipitotemporal cortex.

to quantify the degree of correspondence between a CNN model and the response patterns found in a human visual area.

The leftmost panel of Fig. 2 shows the average correspondence between the CNN models and the RSA matrix found in each participant's visual area when considering all viewing conditions combined. It can be seen that blur-trained CNNs significantly outperformed standard CNNs at predicting cortical responses in early visual areas (V1-V4) as well as high-level object-sensitive areas (LOT, VOT). Moreover, strong-blur CNNs outperformed weak-blur CNNs in V1 through V4, suggesting that stronger levels of blur training may be particularly beneficial for CNN models to better account for neural responses in early visual areas.

An analysis of individual viewing conditions revealed a similar advantage for blur-trained CNNs in predicting fMRI responses in the early visual cortex to clear images and to high-pass filtered images, although no differences were noted for the low-pass filtered condition. Better prediction of cortical responses to high-pass filtered images is notable, as the blur-trained CNNs were not directly trained on high-pass filtered images. This somewhat counterintuitive result was due to the fact that early visual areas exhibited highly confusable responses to the different high-pass filtered object images, whereas standard CNNs excelled at discriminating high-pass filtered stimuli (see Supplementary Fig. 2). By comparison, blur-trained CNNs exhibited more confusable responses to the high-pass filtered objects that led to closer resemblance to human cortical responses.

We next sought to determine whether blur-trained CNNs might show an advantage at predicting the responses of individual neurons recorded from the macaque visual cortex, as single neurons can exhibit far greater stimulus selectivity than is otherwise possible to obtain from fMRI measures of locally averaged neural activity. We first evaluated a popular dataset called BrainScore[48] in which monkeys viewed clear images of objects on natural scene backgrounds while neuronal activity was recorded from areas V1, V2, V4 and inferotemporal cortex (IT). We adopted BrainScore's regression-based approach of using the layer-wise activity patterns of each CNN to fit each neuron's response to a set of training images, and then evaluated its ability to predict responses to independent test images (Supplementary Fig. 3). These analyses revealed that strong-blur CNNs were better able to predict V1 responses than standard CNNs (Fig. 3A, $t(7) = 3.97$, $p = 0.0054$, $d = 1.41$). Moreover, both strong-blur CNNs ($t(7) = 5.35$, $p = 0.0011$, $d = 1.89$) and weak-blur CNNs ($t(7) = 2.97$, $p = 0.0208$, $d = 1.05$) outperformed standard CNNs at predicting neuronal responses in V2. For areas V4 and IT, predictive performance was comparable across standard and blur-trained CNNs.

We also tested CNN performance on another dataset that consisted of neuronal recordings from macaque V1 during the presentation of thousands of natural and synthetic images[49]. In agreement with

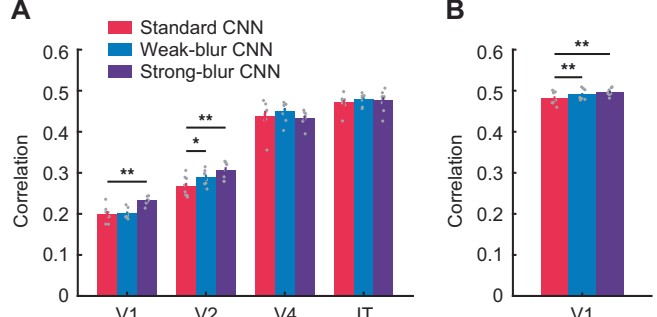

**Fig. 3 | Correlational similarity between CNN model responses and neural responses in macaque visual areas. A** Correlation between predicted and actual neuronal responses in macaque visual areas V1, V2, V4 and IT for regression models based on 8 different standard CNNs (red), weak-blur CNNs (blue) and strong-blur CNNs (purple). The Brain-Score benchmark was employed for data analysis. **B** Correlation between predicted and actual neuronal responses in macaque V1 to thousands of complex images. Error bars indicate ±1 SEM. Gray dots indicate predictive correlation values of individual CNN models. Two-tailed paired t-tests were performed to determine statistical significance (*$p < 0.05$, **$p < 0.01$, and ***$p < 0.001$, uncorrected; the exact p values and raw values are provided in the Source Data).

our findings above, we found that both strong-blur CNNs ($t(7) = 4.53$, $p = 0.0027$, $d = 1.60$) and weak-blur CNNs ($t(7) = 4.65$, $p = 0.0024$, $d = 1.64$) showed better neural predictivity for area V1 than standard CNNs (Fig. 3B and Supplementary Fig. 4). Across both studies, we find that blur-trained CNNs are better able to predict neuronal responses in early visual areas such as V1 and V2. These findings are noteworthy given that the monkeys were tested with clear images only, implying that CNNs that are trained on an exclusive diet of clear images acquire learned representations that deviate from biological vision.

**Visual tuning properties of standard and blur-trained CNNs**

How might training a CNN with a combination of blurry and clear images modify its visual tuning properties, such that it can better account for neural responses in the visual cortex? To address this question, we presented oriented gratings of varying spatial frequency to each CNN and determined which spatial frequencies led to the strongest responses for each convolutional unit in a given layer. This analysis revealed that standard CNNs prefer a much higher range of spatial frequencies, whereas weak-blur CNNs prefer intermediate spatial frequencies and strong-blur CNNs prefer the lowest range of spatial frequencies (Fig. 4A). We further assessed the bandwidth of spatial frequency tuning, a measure that reflects the range of spatial

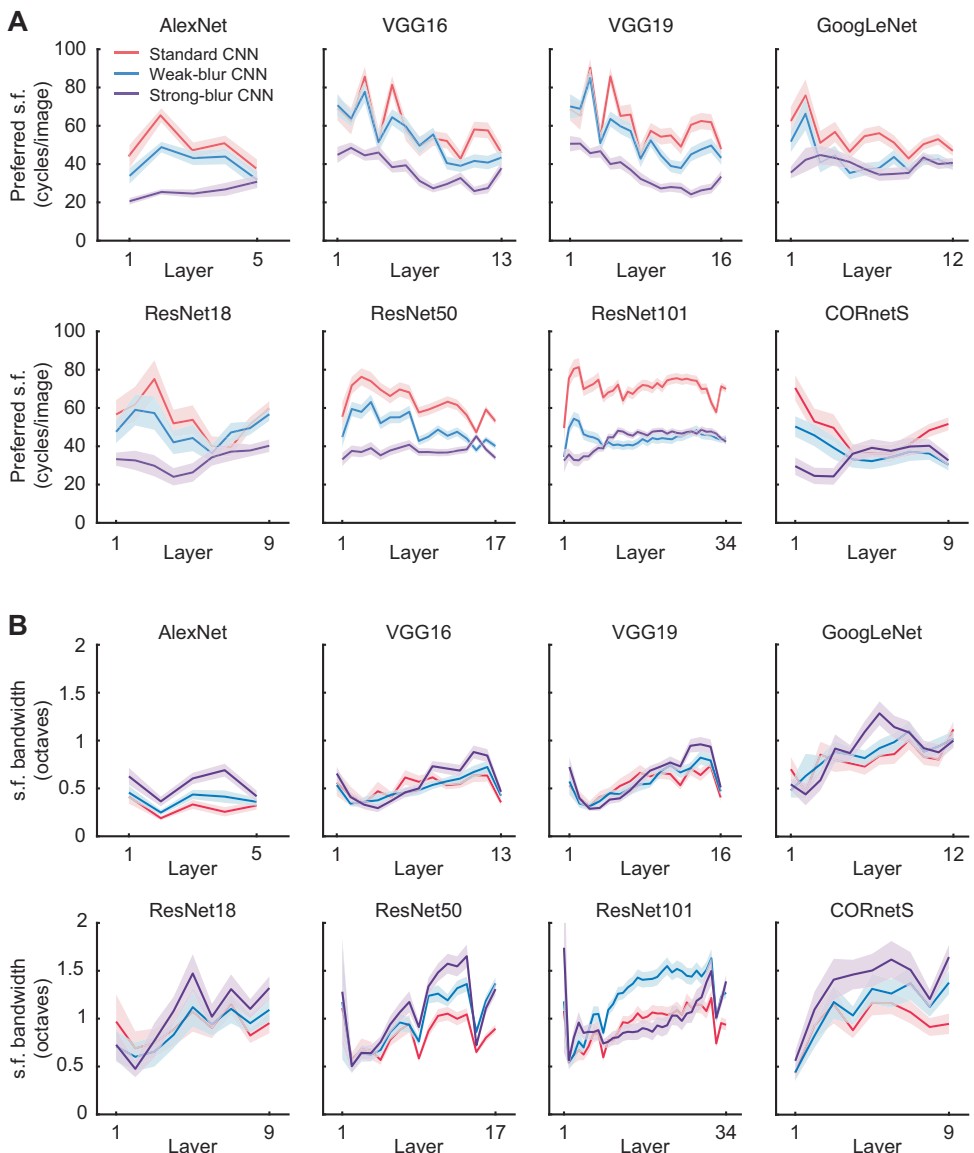

**Fig. 4 | Assessing the spatial frequency tuning of CNNs.** Mean preferred spatial frequency (s.f.) (**A**) and spatial frequency tuning bandwidth (**B**) of individual convolutional units obtained from individual layers of standard CNNs (red), weak-blur CNNs (blue), and strong-blur CNNs (purple). Shaded regions indicate 95% confidence intervals. Source data are provided as a Source Data file.

frequencies for which each unit is tuned. Blur training led to broader spatial frequency tuning bandwidth in most CNNs, particularly in the middle layers (Fig. 4B). Taken together, our findings provide support for the recent proposal that standard CNNs trained on tasks such as ImageNet object classification are heavily biased to emphasize the processing of high spatial frequency for their classification decisions, and are unable to learn or retain the ability to utilize low spatial frequency information for object recognition[6].

Given these shifts in preferred spatial frequency following blur training, we asked whether blur-trained CNNs might exhibit greater sensitivity to object shape information. Although early studies suggested that standard CNNs do show some evidence of shape selectivity[22], subsequent work has revealed that CNNs rely more on textural information than global shape in their classification of hybrid object images[50,51]. Two examples of such hybrid images are shown in Fig. 5B (left), which depicts the global shape of one object filled-in with the texture of a different object. As expected, standard CNNs were strongly biased to classify these hybrid images based on their textural cues, whereas weak-blur CNNs showed a small but highly consistent

shift in favor of shape processing for all 16 object categories that were evaluated (Fig. 5A). These findings concur with a recent study that reported a similarly modest shift in shape sensitivity after a CNN was trained on a combination of clear and blurry images[52]. By comparison, our strong-blur CNNs exhibited a far more pronounced increase in shape bias, and while these networks did not reach human levels of shape bias (gray diamonds), the gap between human performance and CNN model responses was considerably reduced by strong blur training. These findings demonstrate that training CNNs with a subset of highly blurred images can strongly shift their tuning in favor of lower spatial frequency shape information, such that the CNN responses are better aligned with those of human observers.

In addition to quantifying the degree of shape bias exhibited by the CNNs' classification responses, we visualized the image components that the CNNs tended to weigh more heavily for their decisions. We used layer-wise relevance propagation to visualize which features contributed most to the CNN's classification response by decomposing the prediction score backward onto pixel space[53]. Figure 5B shows two examples of texture-shape hybrid stimuli and their layer-wise

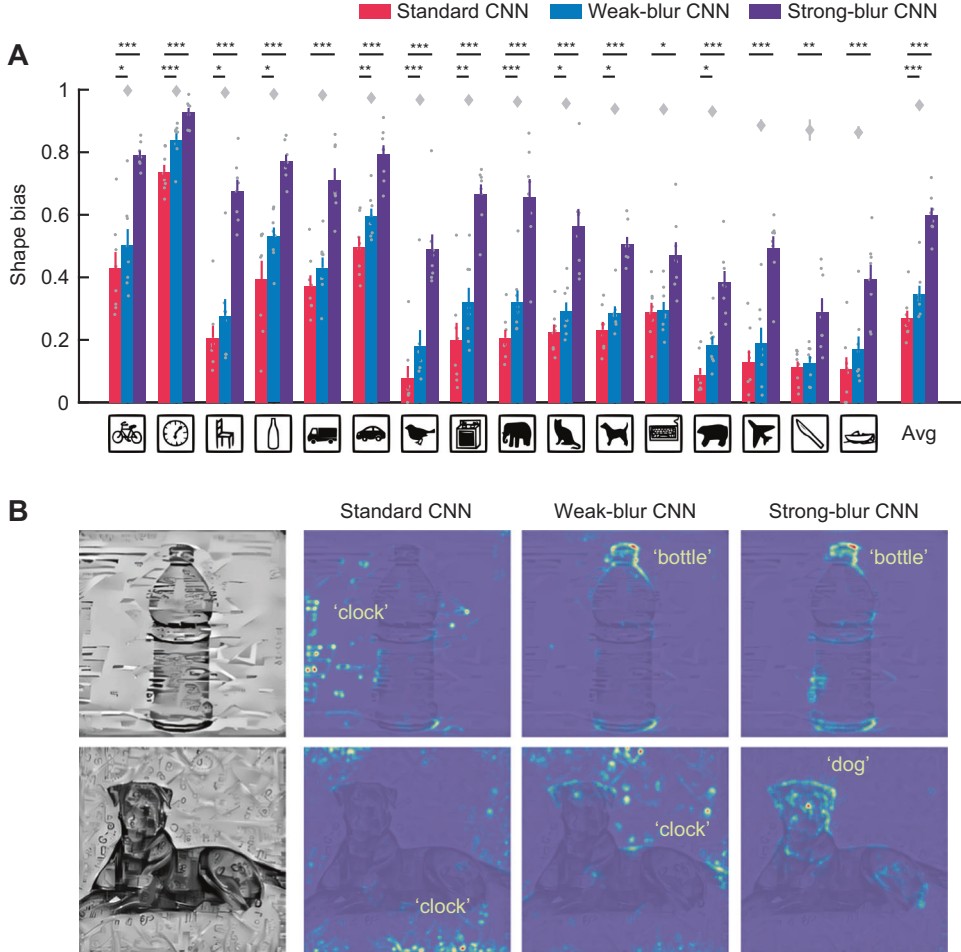

**Fig. 5 | Evaluating the shape bias of CNNs. A** Proportion of shape vs. shape-plus-texture classifications made by standard (red), weak-blur (blue) and strong-blur (purple) CNNs (8 per training condition) when tested with cue-conflict stimuli. Icons indicate category of shape cue tested and bar plots on far right show average shape bias across all 16 categories. Error bars indicate ±1 SEM. Gray dots indicate shape bias score of individual CNN models. Two-tailed paired t-tests were performed to determine statistical significance (*$p < 0.05$, **$p < 0.01$, and ***$p < 0.001$, uncorrected; the exact p values and raw values are provided in the Source Data). **B** Two examples of cue-conflict stimuli (bottle or dog shape with clock texture) from Geirhos et al., 2019 (with permission), shown with corresponding layerwise relevance propagation maps depicting the image regions that were heavily weighted by VGG-19 in determining its classification response.

relevance propagation maps. Whereas standard CNNs tended to emphasize multiple small image patches corresponding to the texture cues that were scattered throughout the hybrid image, the strong-blur CNNs assigned greater weight to coherent diagnostic portions of the primary object, such as the bottlecap on a bottle or the head region of a dog.

### Generalization to challenging out-of-distribution viewing conditions

Given that standard CNNs are strongly influenced by high spatial frequency textural information, might this account for their unusual susceptibility to visual noise[5,30,31]? In a recent behavioral and fMRI study, we found that standard CNNs not only fail to recognize objects in moderate levels of noise, but they also fail to capture the representational structure of human visual cortical responses to objects embedded in noise[5]. Here, we compared standard and blur-trained CNNs in terms of their ability to predict human neural responses to clear objects and those same objects presented in either pixelated Gaussian noise or Fourier phase-scrambled noise. Examples of such stimuli can be seen in Fig. 6 (top row). To do so, we again performed representational similarity analysis on the patterns of fMRI responses in each visual area of interest and each layer of a given CNN (Supplementary Fig. 5).

The benefits of blur training were most evident for the strong-blur CNNs, which outperformed standard CNNs at predicting human cortical responses in both early visual areas and high-level object-sensitive areas when all viewing conditions were analyzed together (Fig. 6, left panel). Focused analyses on fMRI responses to clear objects also revealed better performance for strong-blur than standard CNNs in early visual areas V1-V3, corroborating our earlier findings (Figs. 2 and 3). The strong-blur CNNs performed particularly well at accounting for cortical responses to objects in pixelated Gaussian noise, with improved neural predictivity found across low-level and high-level visual areas. However, strong-blur CNNs were also better at predicting neural responses in early visual areas (V1-V4) to objects embedded in Fourier-phase scrambled noise (sometimes called *pink* noise); such structured noise patterns differ greatly from Gaussian white noise as their power spectrum matches that of natural images. Taken together, we find that blur-trained CNNs can better account for human cortical responses to challenging out-of-distribution conditions involving multiple forms of visual noise. These results provide compelling evidence that blur-trained CNNs provide a better neurocomputational model of the robustness of the human visual system.

Given that blur-trained CNNs showed better prediction of visual cortical responses to clear, blurry, high-pass filtered, and noisy object images, we were motivated to compare both standard and blur-trained

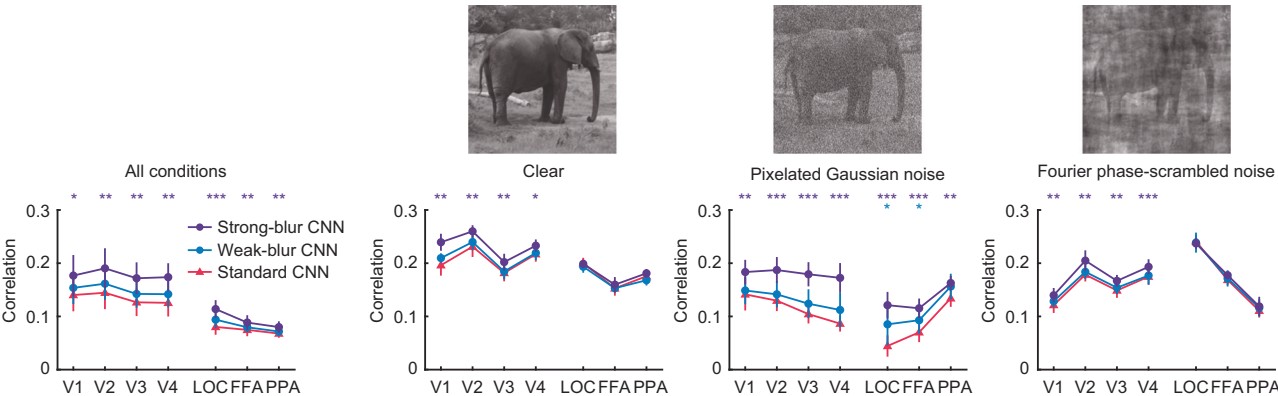

**Fig. 6 | Correlational similarity between CNN model responses ($n = 8$) and neural responses in individual human visual areas ($n = 8$) to clear objects and objects in visual noise.** Leftmost panel shows results pooled across all stimulus conditions; subsequent panels show results for clear objects, objects in Gaussian noise and objects in Fourier phase-scrambled noise; examples of these stimuli are shown above. Error bars indicate ±1 SEM. Two-tailed paired t-tests were performed to determine statistical significance (*$p < 0.05$, **$p < 0.01$, and ***$p < 0.001$, uncorrected; the exact p values and raw values are provided in the Source Data). Elephant image from the authors. LOT, lateral occipitotemporal cortex; VOT, ventral occipitotemporal cortex. LOC, lateral occipital cortex; FFA, fusiform face area; PPA, parahippocampal place area.

CNN models on their ability to deal with a variety of forms of image degradation by employing a popular benchmark, ImageNet-C[54] (https://github.com/hendrycks/robustness). This benchmark consists of the 1000 object categories from ImageNet's validation dataset presented with 19 different types of image degradation (Fig. 7A). Figure 7B shows the impact of image degradation on CNN classification accuracy with noise strength varying from 1 to 5 (i.e., weakest to strongest). We found that blurry image training proved highly effective at improving the robustness of CNNs to most forms of image degradation. Indeed, we observed a significant improvement in performance for 14/19 noise conditions ($p < 0.05$). Weak-blur CNNs showed a consistent increase in classification accuracy for all noise types when compared with standard CNNs, while strong-blur CNNs showed an even greater advantage in many conditions. Specifically, strong-blur CNNs exhibited much greater robustness to both Gaussian blur and other forms of blur (i.e., defocus, glass, motion, zoom). Moreover, strong-blur CNNs were far more robust to all types of pixel-based noise, including Gaussian, speckle, impulse and shot noise. We further found that strong-blur CNNs are more robust to artificial types of image degradation that are known to alter the local image structure of digital images (e.g., elastic transform, JPEG compression, and pixelate). Our findings run contrary to recent claims that CNNs trained on one form of image degradation are unable to generalize to other forms of image degradation[31]. However, strong blur training was not effective at improving robustness to manipulations involving contrast reduction, saturation, spatter or weather-related forms of noise (e.g., Brightness, Fog, Frost, and Snow). Thus, blur training leads to enhanced robustness to many though not all forms of image degradation.

Given that our blur-trained CNNs proved more robust to many forms of randomly generated noise, we sought to test whether they might also exhibit greater robustness to adversarial noise. Adversarial noise involves modifying the pixel values of an original object image in a purposefully deceptive manner designed to shift the CNN's decision to an incorrect object category; even very modest levels of noise that are almost imperceptible to humans can lead CNNs astray[55,56]. We evaluated the adversarial robustness of each CNN by utilizing Projected Gradient Descent[57] with $L_\infty$ and $L_2$ norm constraints ($\epsilon = 0.001$ and 1, respectively). Although blur-trained CNNs remained susceptible to adversarial noise, we found that strong-blur CNNs outperformed standard CNNs with $L_\infty$ of $\epsilon = 0.001$ (19.63% vs.13.61%, t(7) = 7.92, $p = 0.0001$, $d = 2.88$), and both strong-blur (12.04%) and weak-blur CNNs (7.14%) outperformed standard CNNs (4.47%) with $L_2$ of $\epsilon = 1$ (t(7) = 12.45, $p < 10^{-5}$, $d = 4.44$ and t(7) = 4.45, $p = 0.0030$, $d = 1.58$,

respectively). Thus, blur training confers greater robustness to both randomly generated noise and adversarial noise.

## Correspondence with human behavioral responses to out-of-distribution data

We further sought to determine whether blur-trained CNNs might provide a better account of human behavioral responses to challenging out-of-distribution conditions by leveraging a toolbox developed by Geirhos et al. (2021). This toolbox allows for AI models to be compared with human performance on 17 different object recognition tasks, which include multiple forms of image stylization, image modification (e.g., rotation, grayscale conversion), visual noise, as well as high-pass and low-pass filtering[58]. Output measures include overall classification accuracy (called out-of-distribution accuracy), human-AI differences in absolute accuracy, as well as measures of the consistency or agreement between human and AI responses. This analysis revealed that blur training not only improved the out-of-distribution accuracy of CNNs (Fig. 8A), it also led to improved consistency between human and AI responses (Fig. 8BD). For individual CNNs, improvements in human-AI agreement were most prevalent for strong-blur CNNs, followed by weak-blur CNNs, with standard CNNs performing the most poorly. These results demonstrate that blur training improves CNN correspondence with human vision, encompassing human behavioral performance across diverse image conditions.

## Evaluation of recurrent network CORnet-S

We performed a further set of analyses to evaluate whether recurrent visual processing might lead to improved neural predictivity or increased robustness in blur-trained CNNs. Recent studies have found that CORnet-S[59], which performs within-layer recurrent computations in its first 4 convolutional blocks, provides better predictions of neuronal responses in the monkey visual cortex than most other CNN models[27,48]. We compared the performance of CORnet-S with two control networks, one that matched the number of convolutional and fully connected layers of CORnet-S but lacked recurrent processing (CORnet-Shallow) and another feedforward CNN with additional convolutional blocks to match the number of feedforward and recurrent block operations performed by CORnet-S (CORnet-Deep). Our analyses revealed pronounced differences in neural predictivity between these CNNs in the studies that presented low-pass and high-pass filtered images as well as objects in visual noise (see Supplementary Fig. 6). Specifically, blur training was much more beneficial for CORnet-Deep and CORnet-S in comparison to CORnet-Shallow. Likewise,

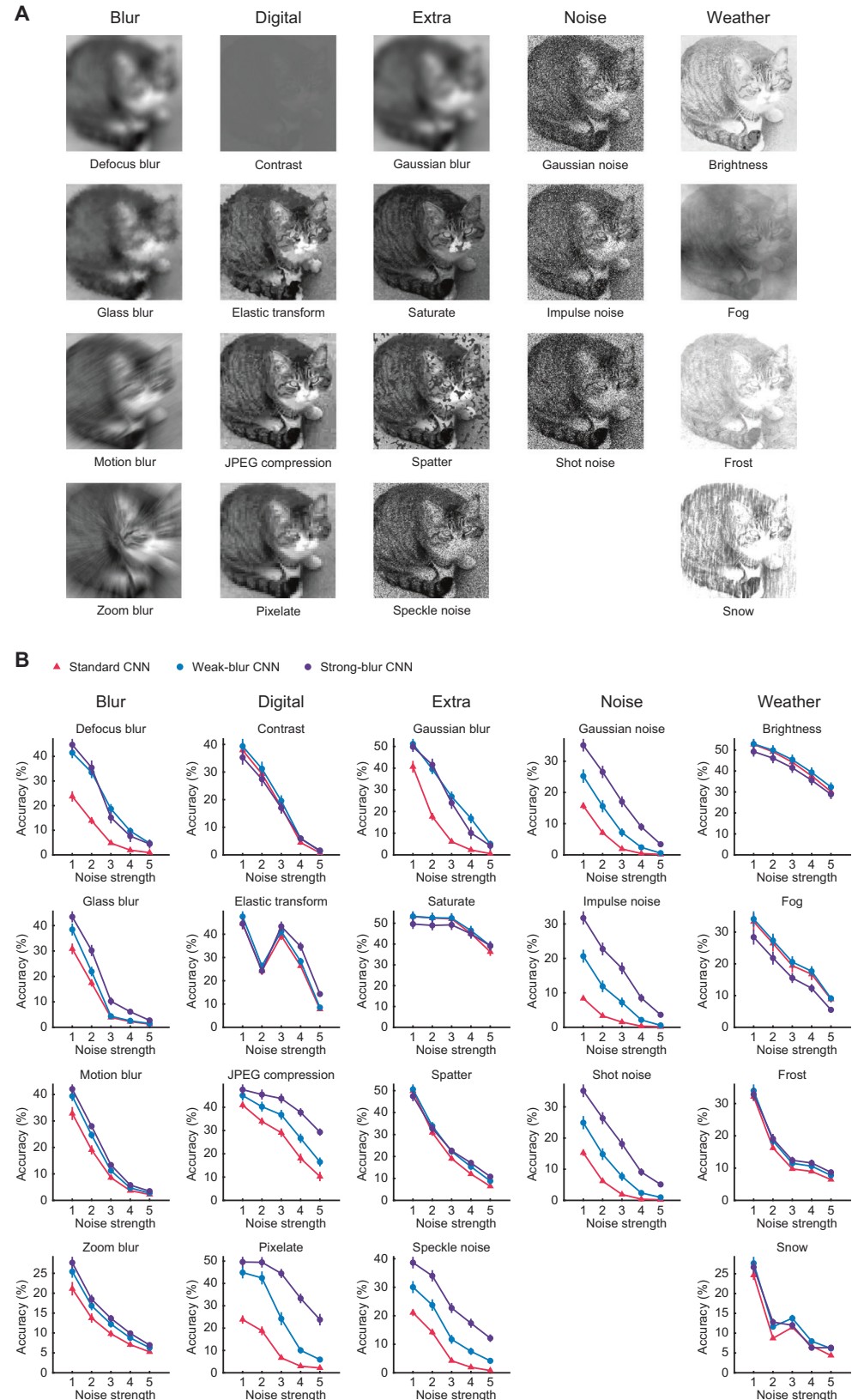

**Fig. 7 | Comparison of CNN robustness to multiple forms of image degradation.** **A** Examples of 19 types of image degradation used by benchmark ImageNet-C to evaluate the robustness of CNNs. Original cat image obtained from https://www.flickr.com/photos/28481088@N00/519609490 and licensed under CC BY 2.0 (with permission from the copyright owner), from which image distorted versions shown here were generated by Hendrycks & Dietterich, 2019. **B** Mean classification accuracy of 8 different standard (red), weak-blur (blue) and strong-blur (purple) CNNs plotted as a function of noise strength for the 19 types of image degradation. Error bars indicate ±1 SEM. Source data are provided as a Source Data file.

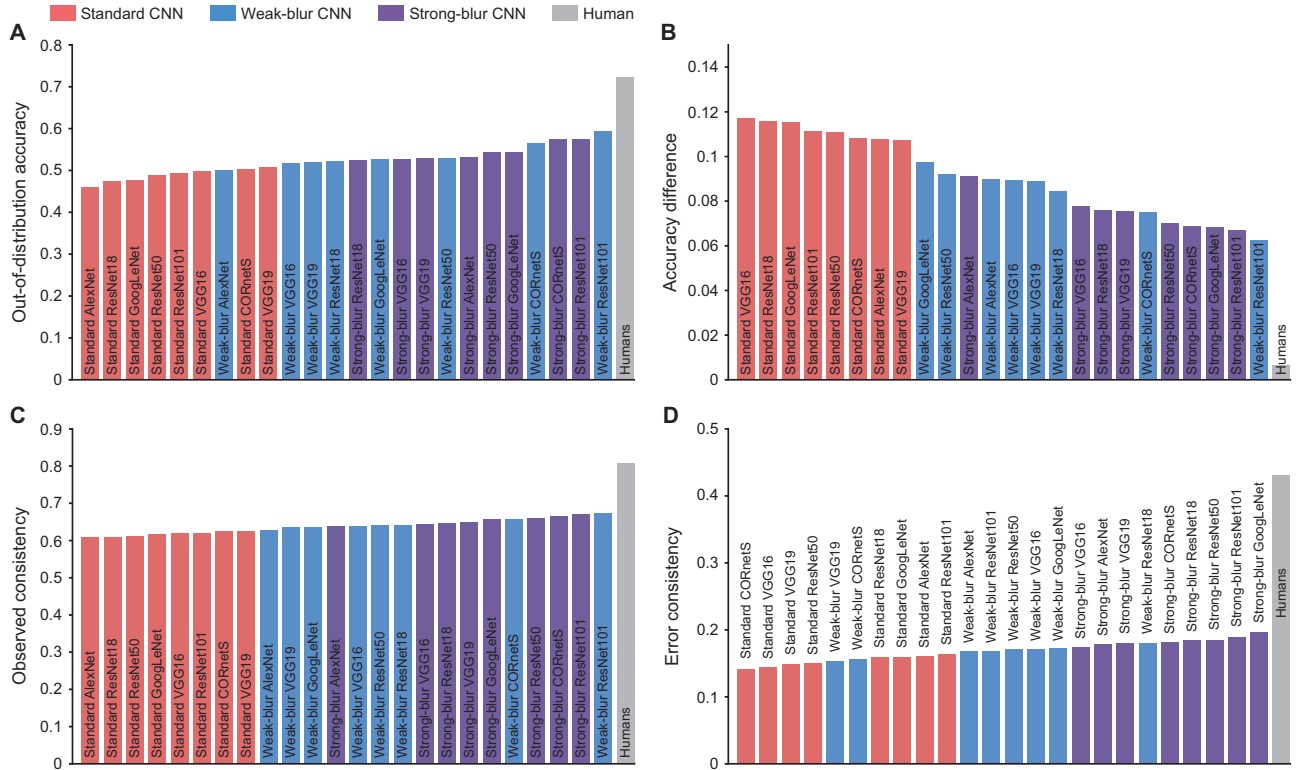

**Fig. 8 | Alignment between human and CNN responses in out-of-distribution scenarios. A** Classification accuracy for standard (red), weak-blur (blue) and strong-blur CNNs (purple) based on aggregated performance for 17 out-of-distribution datasets provided by Geirhos et al. (2021). Note that 1 of 17 conditions involved blurry images, which was not out-of-distribution for the blur-trained CNNs. **B** Accuracy difference between humans and CNN models. **C**, **D** Consistency of responses and error responses between humans and CNNs; higher values indicate better human-AI alignment, with gray bars indicating human-to-human consistency. Source data are provided as a Source Data file.

strong blur training led to the highest levels of overall robustness to image degradation (i.e., ImageNet-C) and also led to greater shape bias for both CORnet-Deep and recurrent CORnet-S, with negligible difference in performance between the latter two CNNs. Our findings indicate that increased model complexity allows a CNN to acquire greater benefits from blur training but we find no additional advantage for recurrent processing over that of feedforward processing. These findings are in general agreement with the fact that the other CNN architectures, excluding CORnet-S, exhibited similarly large benefits from blur training.

**Impact of blur training on visual transformer model ViT**
Finally, we asked whether blur training would necessarily lead to improved robustness, shape sensitivity, or neural predictivity, if applied to a deep neural network model with an entirely different architecture. Whereas CNNs perform filtering and pooling operations designed to mimic the visual system, visual transformer models (ViT) process the information contained in local image patches and the relational information between combinatorial pairs of patch representations through a series of iterative computations[60]. While ViT models operate in a manner that deviates from biological visual systems, they can nevertheless achieve state-of-the-art performance on object classification tasks. We performed standard, weak blur, or strong blur training on 3 ViT models and then evaluated their performance. Blur training led to prominent trends of improved prediction of human fMRI responses, increased shape bias, and greater overall robustness to ImageNet-C, although it did not lead to better prediction of single-unit responses in the monkey (Supplementary Fig. 7). Given that ViT models are not considered to be particularly biologically plausible, the improvements in shape sensitivity and robustness to image degradation are of greater interest here. Thus, even though ViT

models are believed to excel at extracting spatial-relational information, blur training still appears effective at improving their sensitivity to shape.

## Discussion
In this study, we rigorously compared standard versus blur-trained CNNs on their ability to account for neural responses in the visual cortex by leveraging multiple datasets obtained from both monkeys and humans. We reasoned that the existing gap between CNN models and biological visual systems[5,19,22,25,31,47,50,61,62] may be ascribed, at least in part, to inadequate diversity in the set of images that are commonly used to train CNNs. In particular, we hypothesized that blur may be a critical property of natural vision[34,44,45] that contributes to the development and maintenance of robust visual systems. Although we and others have previously posited that exposure to blurry visual input may have the potential to confer some robustness to biological or artificial visual systems, the evidence to support this notion so far has been mixed[6,52,63–65].

Our study reveals that blur-trained CNNs provide a much better neurocomputational model of the visual system's responses to diverse sets of object images. Across multiple neural datasets, we found that blur-trained CNNs outperform standard CNNs at predicting neural responses to clear images in the early visual areas of monkeys[48,49] and human observers[47,66]. Blur-trained CNNs also showed superior neural predictivity for out-of-distribution conditions, including high-pass filtering, objects in pixelated Gaussian noise, and objects in Fourier phase-scrambled noise. Moreover, when we compared CNN versus human performance on a large number of out-of-distribution image datasets[58], blur-trained CNNs consistently outperformed standard CNNs in terms of their ability to account for human behavior. Thus, by incorporating blurry images into the visual diet of CNNs, we can

construct computational models that are better aligned with biological visual systems across a wide range of viewing conditions including those involving visual noise.

This improved robustness to noise is striking given that most state-of-the-art CNNs are severely impaired when Gaussian or other forms of visual noise are added to an object image[5,30–32]. Moreover, it has been reported that if a CNN is trained on one form of visual noise (e.g., Gaussian), one typically observes negligible benefit if it is subsequently tested with a different type of noise (e.g., salt-and-pepper noise)[31] (but see also[5,67]). Here, we evaluated whether blur training might lead to a more generalized improvement in robustness by evaluating the performance of standard and blur-trained CNNs on ImageNet-C[54]. We found that Gaussian blur-trained CNNs can successfully generalize to multiple forms of blur and visual noise, as well as various forms of image compression. That said, blur training did not lead to improved robustness across all conditions; in particular those involving the simulation of noisy weather conditions remained visually challenging. Nevertheless, our findings demonstrate the efficacy of blur training for improving the robustness of CNNs to many forms of image degradation in addition to enhancing their neural predictivity.

How does blur training modify the visual representations learned by CNNs, such that they become both more robust and better aligned with the human visual system? We believe that one key factor is the shift in spatial frequency tuning to favor the processing of lower frequencies and coarser visual features. Another possible contributing factor could be the expanded frequency tuning bandwidth that arose after blur training. Excessive sensitivity to high spatial frequency information appears to be related to a CNN's susceptibility to adversarial noise[68] as well as its ability to learn arbitrary mappings from image datasets with randomly shuffled labels[69]. Thus, the way that standard CNNs process high spatial frequency information seems to deviate considerably from human vision.

In recent work, we have shown that if CNNs are trained on ImageNet object classification with a series of images that gradually progresses from blurry to clear, the CNNs can initially discriminate blurry objects but this ability is quickly lost as they learn to leverage higher spatial frequency information to attain superior classification performance with clearer object images[6]. Such catastrophic forgetting of how to recognize blurry objects clearly deviates from our own visual abilities. Moreover, the image datasets that are commonly used to train CNNs lack the diversity of biological vision as they consist almost entirely of clearly photographed images. Here, by introducing blurry images throughout the training regime of CNNs, the networks must both learn and retain their ability to utilize lower spatial frequency information in order to recognize objects.

Related to this increased sensitivity to low spatial frequency information, we found that blur-trained CNNs become more sensitive to global shape and less sensitive to texture. Several recent studies have suggested that CNNs trained on standard tasks of object classification are unduly influenced by high spatial frequency textural information[6,50–52,65,68,69]. For example, when CNNs are presented with cue conflict stimuli that consist of the global shape of one object and the textural properties of another, their classification decisions are strongly biased by the texture cues[50,51]. Our findings with weak-blur CNNs concur with another recent study, which found that training with moderate levels of blur can lead to a modest increase in shape bias, while the gap between CNN and human shape preference remains large[52]. Here, we found that strong-blur CNNs exhibited a far greater degree of shape bias than standard or weak-blur CNNs, such that they were predisposed to classify the cue conflict stimuli according to their shape rather than their texture over 60% of the time. While blur training alone may not be sufficient to induce the degree of shape sensitivity exhibited by human observers, it does appear to help substantially narrow the gap between artificial and biological vision.

Other methods to increase the shape sensitivity of CNNs have also been proposed. For example, large numbers of hybrid shape-texture conflict stimuli can be generated using style transfer methods[70] so that CNNs can be directly trained to categorize these cue-conflict stimuli according to their shape[51]. Another approach is to train CNNs to become more robust to adversarial noise, which can also improve shape sensitivity and decrease texture bias[71]. Interestingly, a recent study found that CNNs trained with adversarial noise show shifted tuning in favor of lower spatial frequencies in a manner that seems to better match the spatial frequency preferences of V1 neurons[68].

While training with such artificially generated stimuli can improve the shape sensitivity of CNNs, it is not clear how these contrived methods can explain how the human visual system acquires robust, shape-sensitive object representations. Also, although humans do encounter some forms of natural visual noise on occasion (e.g., snow, rain, dust storm), the pervasiveness of blur in everyday vision leads us to posit that blur likely has a primary role in bolstering the robustness of the human visual system.

One might further ask whether the non-uniform application of blur, say to simulate the lower spatial resolution of peripheral vision, might lead to similar improvements in robustness and neural predictivity. Motivated by this question, we conducted an exploratory analysis by training AlexNet on a mixture of clear images and images with progressively stronger blur applied to the periphery (see Methods). The model was then evaluated while withholding the application of peripheral blur. We found that peripheral-blur-trained AlexNet showed much better prediction of human fMRI responses (Supplementary Fig. 8A, B), enhanced shape bias (E), and improved robustness to image degradation (F), and also appeared to show some improvement over clear-trained AlexNet at accounting for neuronal responses in macaque V2, V4 and IT (C). (Previous studies that have explored the impact of peripheral blur training have reported more limited benefits[72], though it can be difficult to compare methodology and findings across studies.) While we are cautious about interpreting the potential neuroscientific implications of these findings, as multiple computational approaches could potentially be adopted to approximate the lower spatial resolution of human vision in the periphery, these findings indicate that multiple options for blur training can be successfully adopted to improve the robustness, shape sensitivity, and neural predictivity of CNNs.

Our results have important implications for both current and future deep learning models of human vision. While considerations such as network architecture and the objective learning function are certainly important for developing more realistic neural network models of the visual system, we propose that the property of blur is likely to be a critical training ingredient for any neural network to learn human-aligned representations of the visual world. Moreover, our findings are not only relevant to the development of better neurocomputational models of the visual system, they may also inform the development of future computer vision applications that must operate in challenging real-world settings. Indeed, by simply incorporating a subset of blurry images into a CNN's training regime, one can attain superior robustness, enhanced shape sensitivity, and much better human-AI alignment with minimal downsides in performance. A variety of image augmentations have been proposed to help bolster the performance of CNNs, including some that have become routine (e.g., random cropping and flipping) and others that are more exotic[73]. Based on our findings, we believe it would be suitable to recommend incorporating blur as a standard form of image augmentation for most computer vision applications. Along these lines, our CNN training code and the weights of our trained networks can be found on a publicly available website with links provided herein.

## Methods

### Training of convolutional neural networks

We evaluated the impact of blur training on 8 CNN architectures implemented in PyTorch: AlexNet[74], VGG16 and VGG19[75], GoogLeNet[76], ResNet18, ResNet50 and ResNet101[77], and CORnet-S[59]. After random initialization, the CNNs were trained to classify 1000 object categories from the training dataset of ImageNet[40] for 70 epochs using stochastic gradient descent with a fixed learning rate of 0.001, momentum of 0.9, and weight decay of 0.0001. Standard CNNs were trained with clear images only, while weak-blur and strong-blur CNNs were trained with a combination of clear and blurry images. All training images were grayscaled, resized to 224 × 224 pixels, randomly rotated by ±10 degrees, and flipped horizontally on 50% of occasions. The images were then normalized using the mean and standard deviation of the pixel intensities of the ImageNet training samples.

For the weak blur condition, the distribution of blur levels was informed by empirical measures of defocus blur that were obtained from binocular eye and scene tracking data[34] while observers performed 1 of 4 different everyday tasks (i.e., ordering coffee, making a sandwich, indoor or outdoor walking). By calculating a scene-based stereo-depth map (spanning 10° eccentricity) with concurrent measures of binocular fixation position, it was possible to calculate the depth distance of objects relative to fixation in each video frame. From these data (courtesy of Sprague et al.), we calculated the extent to which a future fixation target would appear blurred relevant to current fixation (i.e., blur circle size) based on diopter measures of relative depth, measures of mean pupil size (~5.8 mm), and simplifying assumptions pertaining to eye size and other factors[78]. A frequency distribution of blur magnitudes was then obtained, with the different tasks weighted according to their estimated frequency based on Sprague et al.'s analysis of the American Time Use Survey (ATUS) from the U.S. Bureau of Labor Statistics. The weighted distribution of blur circle sizes for subsequently foveated targets was then used to inform the application of blur to the training images (224 × 224 pixels) by assuming that the images were photographed using a 35-mm camera with a 54° horizontal field of view. An exponential function was used to obtain a smoother estimate of the distribution of blur magnitudes. We also adopted a Gaussian blur kernel with FWHM matched to the distribution of blur circle diameters, as blur circles do not adequately account for additional sources of blur such as chromatic aberration. This procedure resulted in a preponderance of clear image presentations (69.4% with σ = 0) and frequencies of 21.3%, 6.5%, 2.0%, 0.6% and 0.2% for which the Gaussian blur kernel was set to a sigma value of 1, 2, 3, 4 or 5 pixels, respectively. It should be noted our assumption of photo zoom size was fairly conservative; if certain training images were taken with a more zoomed-in view (e.g., 50–105 mm), then a greater level of blur would need to be applied to simulate defocus blur for that image.

For the strong blur condition, we presented images at various blur levels with equal frequency (see Fig. 1) based on the fact that the visual resolution steadily declines as a function of eccentricity or distance from the fovea[35,46]. Thus, different levels of resolution remain continually present during natural vision. In addition to clear images, we presented images with Gaussian blur kernels of increasing size (σ = 1, 2, 4, or 8 pixels) to approximate how visual resolution declines from the fovea to the mid-periphery. With the largest blur kernel, the spatial frequency content of the training images (224 x 224 pixels) would be attenuated below 50% amplitude for frequencies exceeding 6 cycles per image, which would impair but not abolish human recognition performance[3,6].

We performed an additional analysis to explore the effect of simulating low-resolution vision in the periphery by applying progressively stronger levels of blur as a function of distance from the center of each training image. To achieve this, we applied a linear increase to the size of the Gaussian blur kernel from the center of the image (coordinates 112, 112 pixel position) to the periphery, starting with a standard deviation of 0 pixels (i.e., clear) at the center and reaching a maximum standard deviation of 8 pixels for eccentricities of 112 pixels or more. We trained AlexNet with a combination of clear and peripheral blur images; the results of which are reported in Supplementary Fig. 8.

### Comparisons between CNN models and human neuroimaging data

We evaluated the correspondence between CNNs and human visual cortical responses by analyzing two publicly available neuroimaging datasets; detailed information can be found in those original papers[5,47]. The first dataset was acquired from 10 observers who viewed clear, high-pass filtered and low-pass filtered images in a 3T MR scanner[47]. Images from 6 different object categories (bodies, cars, chairs, elephants, faces, and houses) were presented using a block paradigm. The high-pass filtered images had a cutoff frequency of 4.40 cycles per degree, while the low-pass filtered images had a cutoff frequency of 0.62 cycles per degree. We analyzed the fMRI data made available for 6 regions of interest: visual areas V1 through V4, lateral occipitotemporal cortex (LOT), and ventral occipitotemporal cortex (VOT). The second dataset was acquired using a 7T MRI scanner from 8 human participants (3 females) while they viewed 16 different clear object images and the same images presented in either pixelated Gaussian noise or Fourier phase-scrambled noise[5]. The object images were selected from 8 object categories (i.e., bear, bison, elephant, hare, jeep, sports car, table lamp, teapot) obtained from the ImageNet validation dataset. The brain regions of interest consisted of visual areas V1 through V4, lateral occipital complex (LOC), fusiform face area (FFA) and parahippocampal place area (PPA).

Representational similarity analysis (RSA) was used to assess the similarity of visual representations across CNN models and human observers. To do so, we calculated the Pearson correlational similarity of the response patterns across all relevant stimulus conditions to obtain a correlation matrix for each visual area of an observer and each layer of a CNN. We could then assess the similarity between human and CNN matrices by calculating their Pearson correlation with the main diagonal excluded. We chose to use Pearson correlation over alternative approaches such as Spearman correlation[79], as the latter allows for non-linear relationships between predicted and actual response patterns that could allow for excessive model flexibility. Moreover, our analyses of the monkey neurophysiology data relied on linear regression; therefore, the use of Pearson correlation to evaluate the human fMRI data seemed more appropriate. Nevertheless, it can be noted that almost identical results were obtained when we applied Spearman correlation instead for our analyses. For the fMRI block paradigm study, the analysis was performed on the mean fMRI response patterns observed for each object category, and on the averaged CNN responses across the 10 images in each object category. For the fMRI objects-in-noise study, the analysis was performed on the mean fMRI responses for each of the 16 object images across the 3 viewing conditions (48 stimulus conditions total).

For the feedforward hierarchical CNNs (e.g., AlexNet, VGG), we performed RSA analysis on every convolutional and fully-connected layer after ReLU non-linearity was applied. For the inception, residual and recurrent networks, we focused our analysis on the layers in which all parallel or recurrent features were combined at the end of each computational block (see Supplementary Table 1).

We calculated the Pearson correlational similarity between each CNN and the response patterns found in a given visual area for each observer and then averaged the results across observers to obtain 8 correlational similarity values (1 per CNN architecture), which allowed us to test for differences in performance between standard, weak-blur and strong-blur CNN training. Statistical tests consisted of repeated measures ANOVA applied across CNN training regimes and visual areas

of interest, followed by planned paired t-tests (two-tailed, uncorrected for multiple comparisons) to directly compare the predictive performance of the different CNN training regimes. For these statistical analyses, all correlation coefficients were first converted to z values using Fisher's r-to-z transformation.

## Comparisons between CNN models and monkey neuronal data

We evaluated the correspondence between CNNs and single-unit responses obtained from the macaque visual cortex by analyzing two publicly available datasets. The first set of analyses focused on data made available (https://github.com/brain-score) as part of the Brain-Score benchmark (http://www.brain-score.org), a site designed to facilitate the evaluation of neural network models and their ability to account for behavioral and neural responses to visual stimuli[48]. We largely adopted the analysis pipeline implemented by Brain-Score to evaluate our CNNs. This involved extracting CNN responses to the object images from each layer, applying PCA to reduce the dimensionality of these responses (to 300 dimensions), and then applying linear partial least squares regression to predict neuronal responses. The Pearson correlation between actual and predicted neuronal responses was calculated using separate sets of images for training and testing, and the median predictivity score across all neurons from a visual area of interest (V1, V2, V4, IT) was then outputted by the Brain-Score toolbox.

Our second set of analyses focused on V1 neuronal data obtained from two alert male monkeys aged 12 and 9 years while they viewed a large set of 7250 natural and synthetic images presented parafoveally[49]. We followed the analysis pipeline of the original study after recoding the analysis in PyTorch. Layerwise CNN responses to the object images were normalized using batch normalization, and a regression model was fitted to the responses of each neuron by using 80% of the images for model training and 20% of the images for model testing. Specifically, a linear/non-linear regression model was trained to minimize a Poission-based loss function via the Adam optimizer. In addition, three regularization constraints were applied to the weights of the regression model: L1-norm sparsity ($\lambda = 0.01$), spatial smoothness ($\lambda = 0.1$), and group sparsity ($\lambda = 0.001$), where $\lambda$ denotes the regularization rate. The correlation between predicted and actual neuronal responses to the independent set of test images was used to evaluate the neural predictivity of the CNN models.

## Spatial frequency tuning preferences of CNN models

We measured the spatial frequency tuning of the convolutional units in each layer of a CNN by presenting whole-field sinusoidal grating patterns that varied in spatial frequency (4.48, 8.96, 13.44,..., 112 cycles/stimulus), orientation (0, 12, ..., 168°), and spatial phases (0, 90, 180, 270°), following previously described methods[6]. The spatial frequency tuning curve was then obtained for individual convolutional units (otherwise known as channels) by averaging the responses across all orientations, phases, and spatial positions. The spatial frequency that elicited the maximum response was identified as the preferred spatial frequency of that unit. We further assessed the bandwidth of spatial frequency tuning by fitting a Gaussian function to spatial frequency response profile of the units on a logarithmic scale, calculating the full width at half maximum, and scaling this value relative to the center frequency of the peak response.

## Texture versus shape bias of CNN models

We evaluated whether CNN classification decisions were more strongly influenced by shape or texture cues by presenting shape-texture cue conflict stimuli that were generated using style transfer methods[70] in the following study[51]. The stimulus set consisted of 1280 images from 16 ImageNet categories that included airplane, bear, bicycle, bird, boat, bottle, car, cat, chair, clock, dog, elephant, keyboard, knife, oven, and truck (available at https://github.com/rgeirhos/texture-vs-shape). For

each hybrid image, the category with the highest confidence response among the 16 categories was identified as the CNN's classification response. The degree of shape bias exhibited by a CNN was then quantified as the proportion of classification decisions that corresponded with the hybrid object's shape in comparison to the total number of shape-consistent and texture-consistent decisions made by that CNN for a given hybrid stimulus set. These CNN results could then be compared with the classification judgments of 10 human participants who were evaluated in the original study.

## Layer-wise relevance propagation

We performed layer-wise relevance propagation to identify the diagnostic features of objects that account for a network's classification decision[53]. This approach works best with strictly hierarchical feedforward CNNs; we therefore focused our analysis efforts by primarily working with VGG-19 using methods and parameter settings we have described elsewhere[5]. To create pixel-wise heatmaps, the relevance score of the unit corresponding to the correct category in the last fully connected layer was set to a value of 1 while all other units were set to 0. Relevance scores were then back-propagated to the input layer to construct heatmaps in pixel space. Only positive values were used to focus on category-relevant features of the target object, and the resulting heatmap was linearly adjusted to a range of 0 to 1.

## Evaluation of adversarial robustness

To evaluate the adversarial robustness of each CNN model, we performed a Projected Gradient Descent-based white-box attack[57]. A key feature of Projected Gradient Descent is its perturbation limit, which controls the extent of input changes. This constraint is vital for ensuring the practicality of the adversarial examples and for setting a uniform standard for comparison, allowing for the evaluation of diverse models under identical conditions. Specifically, this method generates adversarial examples by iteratively updating gradient-based image perturbations with bounded constraints, as formulated by:

$$x_{t+1} = P(x_t + \alpha \bullet sign(\nabla_x L(x_t)))$$

where $x_t$ is the perturbed image at $t$-th step, $P(\bullet)$ is the projection operator to ensure that the adversarial perturbations applied to the image do not exceed a specified threshold, $\alpha$ is the step size, and $L$ is the loss function. The projection operator maps the perturbed image back onto the surface of an $l_p$-norm ball centered at the original image $x$ and bounded by $\|l\|_p \leq \epsilon$. With a random initialization of $x$, the adversarially perturbed data were generated with 15 iterations using a step size of 0.001. We evaluated both $l_\infty$ and $l_2$ norm-bounded perturbations with $\epsilon = 0.001$ and 1, respectively.

## Comparison of CNN outputs and human behavioral responses to out-of-distribution image datasets

We evaluated how closely the outputs of CNNs align with human behavioral responses under out-of-distribution conditions. This comparison was based on publicly available benchmark datasets from Geirhos et al. (2021), encompassing 17 diverse datasets. Twelve of these datasets include parametric variations such as changes in color (both color and grayscale), contrast level, high-pass and low-pass filtering, phase noise, power equalization, opponent color processing, rotation, and three types of Eidolon transformations (I, II, III), as well as uniform noise. The other five datasets focus on nonparametric image alterations, including sketches, stylized, edge, silhouettes, and texture-shape cue conflict. The assessment extends beyond simple accuracy measurements, incorporating three additional metrics: 1) the accuracy difference, which compares CNN and human accuracy across various out-of-distribution tests; 2) observed consistency, which measures the proportion of instances where both humans and a CNN model either correctly or incorrectly identified the same sample; and 3) error

consistency, which examines whether the observed consistency exceeds that of two independent decision-makers with similar accuracy levels. The relevant code and datasets are accessible online at https://github.com/bethgelab/model-vs-human.

## Reporting summary
Further information on research design is available in the Nature Portfolio Reporting Summary linked to this article.

## Data availability
All datasets employed in this study are open to public access and are sourced from their original publications, i.e., two fMRI datasets (https://osf.io/tsz47/, https://osf.io/bxr2v/), BrainScore (https://github.com/brain-score), Macaque V1 data (https://doi.org/10.12751/g-node.2e31e3/), ImageNet-C (https://github.com/hendrycks/robustness), and Model-vs-human (https://github.com/bethgelab/model-vs-human). All of the source data used to create the figures in this paper are available as a Source Data file. Source data are provided with this paper.

## Code availability
The codes used for training and visualization, along with the trained weights for individual CNN models, are available on the Open Science Framework at https://osf.io/upf5w/. The blur augmentation technique used in this study is implemented using Python 3.7 with the PyTorch library version 1.13.1 and the Kornia library version 0.5.8. The details are described on our GitHub page at https://github.com/hojin89/BlurTraining (https://doi.org/10.5281/zenodo.10468454).

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

## Acknowledgements

We thank Bill Sprague, Emily Cooper and Martin Banks for sharing their depth map data from 4 natural viewing tasks to inform our weak-blur training condition. This research was supported by the following grants from the National Eye Institute, National Institutes of Health (NEI/NIH):

R01EY035157 and R01EY029278 to FT, and P30EY008126 to the Vanderbilt Vision Research Center.

## Author contributions

H.J. and F.T. conceived of and designed the study, H.J. trained all relevant CNN models and performed all data analyses, H.J. and F.T. wrote the manuscript together.

## Competing interests

The authors declare no competing interests.
