## [Peer Review File · Nature Communications]

Improved modeling of human vision by incorporating robustness to blur in convolutional neural networksReviewer #1 (Remarks to the Author):

Key results

This study revealed that training with blurred images helps convolutional neural networks (CNNs) acquire visual representations more similar to biological brains than standard training with only clear images. Moreover, these blur-trained CNNs showed higher robustness to out-of-distribution distortions and higher shape bias than standard CNNs. These results implicate that simulating our experience with blurry retinal images in everyday life is one of key elements for building a computational model of the biological visual system.

Significance

Similar in-silico experiments investigating the effect of training with blurred images have been performed in several previous studies, including those of the authors (all of which are cited). Some of the results reported in this study, such as the improved shape bias and shift of frequency tuning, is not completely new (the authors duly acknowledge this). However, this study stands out from those previous studies in that it examines the effects of blur training from a broader perspective, from comparisons with multiple neurophysiological data to an examination of out-of-distribution robustness, and finds stronger support for the hypothesis that the exposure to blurry images underlies robust biological vision.

Clarity and context

This paper is well motivated and very clearly written. Every paragraph reads well and easy to follow.

Reference

I did not find any places where the references were considered inappropriate.

Validity

The authors tested eight different CNN architectures with various depths and with or without recurrence, and found that the results were mostly consistent. Therefore, the results obtained appear to be robust. I found no flaws in the evaluation protocols. The interpretation of the results seems mostly fair (except for the following point).

One thing that I do not understand is why blur training did improve the correlation for high-pass filtered images in the first main result (Fig. 2). Given that the blur-trained CNNs showed tuning shifts in the lower frequency side (Fig. 4), a natural consequence would be to increase the correlation for the low-pass filtered images. Contrary to this expectation, the blur training improved the correlation for the high pass filtered images. The authors only suggested that this could be a sign of out-of-distribution robustness, but this is not convincing enough since the blur-trained CNNs did not show improved correlation for low-pass filtered images.

A possible explanation could be that the early visual areas in the human brain does not represent high-frequency information well enough to discriminate the object categories under test, and that the blur training degraded the high-frequency representation of the CNNs, bringing it somewhat closer to the human representation. This possibility would be testable by computing the representational similarity between the low-pass and high-pass image sets for both human data and CNNs. In addition, it would also be important to see the bandwidth of the frequency tuning of the internal layers of the CNNs. The current results (Fig. 4) only show the peak frequency, but I wonder if the blur training only shifts the tuning or broadens the bandwidth without sacrificing the high-frequency channels.

Data and methodology

All the neurophysiological data (human fMRI responses and monkey neural responses) were taken from the publicly available dataset. The training procedure of CNN models follows standard methodology and is described well in detail to ensure reproducibility.

Analytical approach

I wonder why the authors used Pearson correlation to calculate the similarity between human and CNN RDMs. Pearson correlation assumes a linear relationship between data, but it seems a little

too strict to assume such a relationship between CNN and fMRI RDMs. I am curious about whether the similar patterns of results can be obtained when another similarity measure that does not assume linearity, such as Spearman correlation, is used.

I did not find any problems in the other parts of the analysis. However, as I have no experience in fMRI data analysis by myself, I would like to leave the rigorous review of that part to another reviewer.

Suggested improvements

The paper focuses on the similarity of the internal representations of CNNs and brains, but it is also interesting to see how close the blur trained CNNs are to humans at the behavior level. The current results suggested that the stronger the blur size of images experienced during training, the more similar the representation of the CNNs is to the brain. Could something similar be observed at the level of object recognition performance?

A similar question arises in the results of the out-of-distribution robustness test using ImageNet-C. Here, the authors demonstrated that the blur-trained CNNs exhibit better classification performance than the standard-trained ones for several out-of-distribution distortion types, but without discussing the extent to which humans can classify object categories under these distortions.

As a way to address the above questions, I would suggest using a publicly available toolbox developed to benchmark the similarity between humans and machine learning models:

<https://github.com/bethgelab/model-vs-human>

Reference: Geirhos et al. Partial success in closing the gap between human and machine vision. NeurIPS2021.

The beauty of this toolbox is that it can evaluate image-level error consistency, a finer measure than classification accuracy.

Another point that remains unclear is how special blur distortion is compared to other data augmentations such as Gaussian noise and adversarial noise, both of which are known to produce low-frequency bias (Yin et al., A Fourier Perspective on Model Robustness in Computer Vision, NeurIPS2019). While I understand that the blur-training is more biologically plausible, it would be beneficial for the machine learning community to compare and quantify the effectiveness of blur training with the other types of data augmentation. Although the authors repeatedly emphasize the superiority of the blur training over others that it can lead to better generalization to out-of-distribution performance, it is difficult to prove this without comparing models trained under the same procedure, varying only the augmentation type and strength. In fact, a previous study (Geirhos et al., 2018, Generalisation in humans and deep neural networks), which the authors cited to show the weak generalization performance of other augmentations, also used low-pass filtering for training, but did not observe the similar amount of generalization as presented in the current study. It would be nice if the authors could clarify what is responsible for this difference.

Reviewer #2 (Remarks to the Author):

This manuscript presented very interesting work on the improvement of convolutional neural networks in order to match the neuronal processes of human vision. The strength of the work is that the authors showed the validity of their proposed models from a wide range of systematical evaluations of various datasets and settings from the human fMRI to macaque physiological data, shape vs. texture cue-conflict settings, images with different visual noises and degradation settings, and the models with/without recurrent visual processing. This reviewer considers that the presented study has great potential to open up the development of more human-like computer vision models based on neural networks. However, there are several points to be addressed and clarified before considering this work to be published.

Major comments:

1. One of the main questions I had was whether the computer vision model to mimic human vision can be obtained by simply training the model using images with an object with various blur levels. As the authors mentioned multiple times in the manuscript, the human visual acuity systematically declines from the fovea to the periphery. Therefore, this reviewer thinks that the computer vision model should be trained using the images that the human perceives, such as the image with the high/clear pixel values near the center, and then the pixel values are slowly degraded/blurred from the center to the side.

Can authors show the results from this perspective and compare them with their presented mapping results using the trained CNN model from images with a mixture of clear/blurred images?

2. Also, ideally, it would be most ecologically valid if the CNN model training and particularly testing could be performed from the image with multiple objects in it while the blur level continuously changes from the center to the side by mimicking the visual scene in the retina. To this end, the multiple objects in the images can first be segmented and then recognized in their categories using the trained CNN models.

3. (Paragraph#2, p.4 in Results) "This manipulation was informed by the fact that..."
Related to this point, the corresponding reference seems outdated, so this reviewer is unsure of the validity of the setup of specific target ratios for the frequency of blurry images.

4. (Paragraph#5, p.4 in Results) "Better prediction of cortical responses to high-pass filtered images is notable, as the ..."
Can authors elaborate on this part, as it is not straightforward to understand the reasoning behind it?

5. (Paragraph #2, p.13 in Materials and Methods) "..., $P(\cdot)$ is the projection operator onto the l_p ball around x bounded by $\|l_p\| \leq \epsilon$, ..."
Can authors elaborate on this method about Projected Gradient Descent-based adversarial examples?

6. The reasoning behind how the strong-blur CNN can focus on the informative regions of the object identified from the layerwise relevance propagation scheme is not fully straightforward. For example, shape detection can be advantageous when a high-pass filtered image is used compared to the blurred image. This is also perhaps related to the counter-intuitive results on the better prediction for high-pass filtered images using blur-trained CNNs than standard CNNs. Can authors provide their intuition as to why this was the case?

Minor comments:

7. (Paragraph# 4, p.11 in Materials and Methods) "... and then averaged the results across observers to obtain 8 observations (1 per CNN architecture), ..."
Here, the word, observations must denote the different CNN architecture. If so, it is recommended to use an alternative word since the observer was also used to denote the subject in the same sentence.

8. Also, the correlational similarity metric used must be spelled out in this sentence.

Reviewer #3 (Remarks to the Author):

This article aims to test the hypothesis that ecological levels of image blur (introduced by retinal sampling and optical defocus) could be instrumental in shaping visual cortex representations. To this end, the authors train a number of convolutional neural networks (CNNs) for image classification with normal, weak-blur or strong-blur image datasets. In support of their hypothesis, they find that blur-trained networks show increased correspondence to human and monkey brain representations in early visual cortex, increased robustness to various types of image perturbations (including adversarial attacks), and increased reliance on shape vs. texture

information. These findings are doubly important: they help us understand the role that blur can play in shaping cortical representations; and they help close the gap between machine and biological vision. Therefore, the study should be of interest to both neuroscientists and AI scientists. The paper was very-well written, and the arguments fairly convincing, thus I only have minor suggestions.

1. The fact that blur training improves reliance on shape (compared with texture) information was already the point of a recent paper (Yoshihara et al, Frontiers, 2023). (I emphasize that the other conclusions on representational similarity and noise/adversarial robustness were not part of the previous study). The authors acknowledge this paper, but describe it as similar to the weak-blur conditions, and suggest that their strong-blur condition yields qualitatively different results. I was not very convinced by this line of reasoning, because the blur training regimes are difficult to compare across the two studies, and the qualitative conclusions (increased reliance on shape, but still short of human behavior) hold for both studies and for both weak-blur and strong-blur training (in other words, the difference is only quantitative). I would much prefer presenting this part of the experiment as a replication of Yoshihara et al's main finding, with an additional conclusion, that shape reliance can be increased by increasing blur (to some extent).

2. I appreciated the use of a diverse number of CNNs, including Inception and ResNet-based architectures, as well as a recently proposed recurrent model (CORnetS). However, the state-of-the-art in image classification in recent years has shifted from CNNs to Transformer-based architectures. I was curious (and other readers might be as well) about the possibility of extending the present findings to such architectures? Is it just too computationally expensive to retrain a ViT-like model on blurry images? Or are there theoretical reasons why the authors think the approach would not work? If so, this could be addressed in the Discussion.

3. One implication of the findings (targeted to a computer-vision audience) is that it would be a good idea (and a particularly cheap one) to include blurring as a default augmentation strategy for standard image classification libraries. I'm not suggesting that the authors undertake this themselves, but this could be a direct recommendation emanating from the present study?

REVIEWER COMMENTS

Reviewer #1 (Remarks to the Author):

Key results

This study revealed that training with blurred images helps convolutional neural networks (CNNs) acquire visual representations more similar to biological brains than standard training with only clear images. Moreover, these blur-trained CNNs showed higher robustness to out-of-distribution distortions and higher shape bias than standard CNNs. These results implicate that simulating our experience with blurry retinal images in everyday life is one of key elements for building a computational model of the biological visual system.

Significance

Similar in-silico experiments investigating the effect of training with blurred images have been performed in several previous studies, including those of the authors (all of which are cited). Some of the results reported in this study, such as the improved shape bias and shift of frequency tuning, is not completely new (the authors duly acknowledge this). However, this study stands out from those previous studies in that it examines the effects of blur training from a broader perspective, from comparisons with multiple neurophysiological data to an examination of out-of-distribution robustness, and finds stronger support for the hypothesis that the exposure to blurry images underlies robust biological vision.

Clarity and context

This paper is well motivated and very clearly written. Every paragraph reads well and easy to follow.

We very much appreciate Reviewer 1's comments and encouraging feedback. In writing the manuscript, we did strive to use language to make it accessible to a wider audience.

Reference

I did not find any places where the references were considered inappropriate.

Validity

The authors tested eight different CNN architectures with various depths and with or without recurrence, and found that the results were mostly consistent. Therefore, the results obtained appear to be robust. I found no flaws in the evaluation protocols. The interpretation of the results seems mostly fair (except for the following point).

One thing that I do not understand is why blur training did improve the correlation for high-pass filtered images in the first main result (Fig. 2). Given that the blur-trained CNNs showed tuning shifts in the lower frequency side (Fig. 4), a natural consequence would be to increase the correlation for the low-pass filtered images. Contrary to this expectation, the blur training improved the correlation for the high pass filtered images. The authors only suggested that this could be a sign of out-of-distribution robustness, but this is not convincing enough since the blur-trained CNNs did not show improved correlation for low-pass filtered images.

A possible explanation could be that the early visual areas in the human brain does not represent high-frequency information well enough to discriminate the object categories under test, and that the blur training degraded the high-frequency representation of the CNNs, bringing it somewhat closer to the human representation. This possibility would be testable by computing the representational similarity between the low-pass and high-pass image sets for both human data and CNNs. In addition, it would also be important to see the bandwidth of the frequency tuning of the internal layers of the

CNNs. The current results (Fig. 4) only show the peak frequency, but I wonder if the blur training only shifts the tuning or broadens the bandwidth without sacrificing the high-frequency channels.

We concur with the reviewer's observation that the enhanced correlation observed for fMRI responses to high-pass filtered images deviated from our initial expectations and requires further elucidation. Consistent with the reviewer's insights, we hypothesized that conventional neural networks are excessively biased to process high spatial frequency information for object categorization, a process that diverges from the representational patterns of the human brain.

To investigate this issue in greater detail, we inspected the average representational similarity analysis (RSA) matrices, both for human subjects and CNNs, where we selected the specific CNN layer that most accurately predicted the neural activity of individual human brain regions. As can be seen in the figure below, activity patterns in human visual areas V1-V4 are actually more confusable (i.e., higher off-diagonal correlation values) for high-pass filtered objects than for low-pass filtered objects, whereas standard CNNs exhibit weaker off-diagonal correlations for high-pass filtered objects and much stronger off-diagonal correlations for the low-pass filtered objects. The confusability of the high-pass filtered objects is enhanced in the strong-blur-trained CNNs and the overall RSA pattern of confusability is more similar to that observed in areas V1-V4, which accounts for our observed results. We have added a discussion of the above points to the revised results section (Changes applied to lines 166-171 in the manuscript), and now include **Supplementary Figure 2** of the manuscript to show these RSA results.

Supplementary Figure 2. Plots showing the mean Pearson correlational similarity matrices for human fMRI participants and CNNs in response to the clear, high-pass filtered and low-pass filtered object images used by Xu and Vaziri-Pashkam (2021). The CNN matrices show results for the layer that most accurately predicted neural activity in a specified human visual area.

The reviewer's suggestion to look at possible changes in spatial frequency bandwidth is an interesting one. We performed a follow-up analysis by fitting each individual tuning curve using a Gaussian function on a logarithmic scale and calculating the FWHM of the fitted Gaussian. The bandwidth of the resulting curve was then calculated using the formula $(2 * \text{sqrt}(2 * \log(2)) * \sigma) / \mu$, where μ is the center position of the peak, and σ is the standard deviation of the Gaussian, following the methodology described by David, Hayden, and Gallant (2006). The results indicate a suggestive trend that blur training may lead to broadening of the spatial frequency tuning bandwidth, particularly in the middle and higher layers of the CNNs. We have incorporated these results into **Figure 4B** of the manuscript, where they are presented and discussed alongside with the peak frequency results (Changes applied to lines 205-208, 409-410, and 637-640 in the manuscript).

Figure. Mean bandwidth of the spatial frequency (s.f.) tuning curves for convolutional units from individual layers of standard CNNs (red), weak-blur CNNs (blue), and strong-blur CNNs (purple). Shaded regions indicate 95% confidence intervals.

Data and methodology

All the neurophysiological data (human fMRI responses and monkey neural responses) were taken from the publicly available dataset. The training procedure of CNN models follows standard methodology and is described well in detail to ensure reproducibility.

Analytical approach

I wonder why the authors used Pearson correlation to calculate the similarity between human and CNN RDMs. Pearson correlation assumes a linear relationship between data, but it seems a little too strict to assume such a relationship between CNN and fMRI RDMs. I am curious about whether the similar patterns of results can be obtained when another similarity measure that does not assume linearity, such as Spearman correlation, is used.

We realize that different researchers favor different approaches for comparing and fitting model results to neural data. We have added some text to our revised methods section explaining why we

favor using Pearson correlation for our RSA analyses rather than other approaches such as Spearman correlation (e.g., Kriegeskorte et al., 2008), which does not assume a linear mapping between the predicted and actual responses. That said, we observed very similar results if the same CNN-fMRI analysis is performed using Spearman correlation (see below), and report this fact in the revised methods section (Lines 575-581 in the manuscript).

(Page 13, paragraph #1 in Materials and Methods) "We chose to use Pearson correlation over alternative approaches such as Spearman correlation⁷⁹, as the latter allows for non-linear relationships between predicted and actual response patterns that could allow for excessive model flexibility. Moreover, our analyses of the monkey neurophysiology data relied on linear regression; therefore, the use of Pearson correlation to evaluate the human fMRI data seemed more appropriate. Nevertheless, the pattern of results we observed were almost identical when we applied Spearman correlation instead for our analyses."

Coming from a tradition of early visual processing, our lab favors using a stricter criterion for evaluating the mapping between independent and dependent variables. For example, extensive modeling work has been conducted to describe the contrast-response function observed in cortical areas such as V1; if one were to forego the linear mapping between stimulus contrast and output response, then any model that predicted monotonically increasing activity as a function stimulus strength would be just as good as any other model. We believe that allowing for such non-linear relationships may lead to excess model flexibility, and that in turn could impede one's ability to distinguish between the goodness-of-fit of competing models in certain cases.

Moreover, a substantial portion of our manuscript consists of the evaluation of neuronal data recorded from the macaque monkey. The analytical approach most commonly used in these studies (e.g., BrainScore measures) involves applying a multivariate regression-based approach to map from the CNN model to the observed neuronal responses to each image, assuming a linear mapping. For greater consistency across our analyses of neuronal and fMRI data, we therefore chose to use Pearson correlation for our representational similarity analyses.

Xu and Vaziri-Pashkam (2021)

Jang et al. (2021)

Figure. Spearman correlation-based similarity between CNN model responses and neural responses in different human visual areas to clear, high-pass filtered and low-pass filtered images from Xu and Vaziri-Pashkam (2021) (top) and to clear images, pixelated Gaussian noise, and Fourier phase-scrambled noise from Jang et al. (2021) (bottom). Standard CNNs (red), weak-blur CNNs (blue) and strong-blur CNNs (purple) were evaluated.

I did not find any problems in the other parts of the analysis. However, as I have no experience in fMRI data analysis by myself, I would like to leave the rigorous review of that part to another reviewer.

Suggested improvements

The paper focuses on the similarity of the internal representations of CNNs and brains, but it is also interesting to see how close the blur trained CNNs are to humans at the behavior level. The current results suggested that the stronger the blur size of images experienced during training, the more similar the representation of the CNNs is to the brain. Could something similar be observed at the level of object recognition performance?

A similar question arises in the results of the out-of-distribution robustness test using ImageNet-C. Here, the authors demonstrated that the blur-trained CNNs exhibit better classification performance than the standard-trained ones for several out-of-distribution distortion types, but without discussing the extent to which humans can classify object categories under these distortions.

As a way to address the above questions, I would suggest using a publicly available toolbox developed to benchmark the similarity between humans and machine learning models:

<https://github.com/bethgelab/model-vs-human>

Reference: Geirhos et al. Partial success in closing the gap between human and machine vision. NeurIPS2021.

The beauty of this toolbox is that it can evaluate image-level error consistency, a finer measure than classification accuracy.

We thank the reviewer for this very helpful suggestion. Based on this recommendation, we downloaded the toolbox and evaluated our full set of models, which were trained from scratch using 1000-category ImageNet training with standard, weak blur, or strong blur conditions. As can be seen in the figure below, our blur-trained CNNs consistently outperform CNNs trained using standard clear images with respect to classification accuracy (i.e., labeled out-of-distribution accuracy; note that while low-pass filtered images as a test set are not strictly out-of-distribution, we retain the terminology from Geirhos et al., 2021). Moreover, they show improved consistency and improved error consistency when compared to the human performance data collected by Geirhos et al. (2021). We have added a new section to our results (see *Correspondence with human behavioral responses to out-of-distribution data*; lines 308-322 and 686-702 in the manuscript) that discusses our evaluation of the models using the toolbox. In this section, we introduce a new figure (**Figure 8**) in the revised manuscript, as further evidence that blur training does indeed improve the correspondence between deep neural network models and human vision.

Figure 8. A Classification accuracy for standard (red), weak-blur (blue) and strong-blur CNNs (purple) based on aggregated performance for 17 out-of-distribution datasets provided by Geirhos et al. (2021). Note that 1 of 17 conditions involved blurry images, which was not out-of-distribution for the blur-trained CNNs. **B** Accuracy difference between humans and CNN models. **C, D** Consistency of responses and error responses between humans and CNNs; higher values indicate better human-AI alignment, with gray bars indicating human-to-human consistency.

Another point that remains unclear is how special blur distortion is compared to other data augmentations such as Gaussian noise and adversarial noise, both of which are known to produce low-frequency bias (Yin et al., A Fourier Perspective on Model Robustness in Computer Vision, NeurIPS2019). While I understand that the blur-training is more biologically plausible, it would be beneficial for the machine learning community to compare and quantify the effectiveness of blur training with the other types of data augmentation. Although the authors repeatedly emphasize the superiority of the blur training over others that it can lead to better generalization to out-of-distribution performance, it is difficult to prove this without comparing models trained under the same procedure, varying only the augmentation type and strength. In fact, a previous study (Geirhos et al., 2018, Generalisation in humans and deep neural networks), which the authors cited to show the weak generalization performance of other augmentations, also used low-pass filtering for training, but did not observe the similar amount of generalization as presented in the current study. It would be nice if the authors could clarify what is responsible for this difference.

The reviewer raises a number of interesting points about other options for data augmentation and whether training with Gaussian or adversarial noise might lead to similar improvements in generalization performance as blur training. Our original submission included some discussion of these points, and in our revision, we have expanded our discussion of these issues and related studies. In our recently published work in PLOS Biology (Jang et al., 2021), we found that CNNs trained to recognize objects in Gaussian noise also exhibit successful generalization to other forms of visual noise (e.g., salt and pepper noise) as well as better recognition of low-pass filtered images, in

general agreement with the above-mentioned study by Yin et al. but apparently in contrast to the report by Geirhos et al. While humans certainly do encounter noise elements on occasion (e.g., snow, dust, rain), they encounter visual blur far more often, and for these reasons, we posit that it is more likely that real-world experiences with blur confer some robustness to noise rather than the other way around. However, we acknowledge that it would be very useful for future studies to obtain estimates of the frequency of blur, noise and other forms of image degradation in everyday human vision and characterize the visual statistics of those conditions. We have revised our discussion section to address these issues to a greater extent (Changes applied to lines 396, 451-456 in the manuscript).

The fact that our blur-trained CNNs exhibit successful generalization to out-of-distribution conditions, including the behavioral conditions that Geirhos et al (2021) evaluated, further demonstrates that our blur-trained networks do well at generalizing to other challenging visual conditions such as those involving visual noise. We are highly confident of our model results as we evaluated multiple CNN models in this study and observed consistent improvement in generalization performance for all blur-trained models, including ResNet50 which was the primary model evaluated by Geirhos et al. (2018). Note that multiple aspects of our training protocol differed from Geirhos et al., and it is conceivable that any of these differences (or others we might not be aware of) could account for the lack of generalization that Geirhos et al. (2018) observed. These include the fact that we used separate object images for training and test while Geirhos et al. used the same object images at test (after applying image degradation); also, we trained CNNs with all 1000 object categories in ImageNet while Geirhos et al. used a more limited set of 16 object categories.

While we agree that it would be of interest for future studies to investigate why Geirhos et al. found weak generalization to out-of-distribution conditions, this question falls outside of the main goals of our study, which is to report how blur-training leads to broad-ranging improvements in CNN performance and better alignment with the human visual system. Also, we would be hesitant to directly speculate in our manuscript as to why Geirhos et al. observed effects of DNN training that appear consistent with potential problems of overfitting.

Reviewer #2 (Remarks to the Author):

This manuscript presented very interesting work on the improvement of convolutional neural networks in order to match the neuronal processes of human vision. The strength of the work is that the authors showed the validity of their proposed models from a wide range of systematical evaluations of various datasets and settings from the human fMRI to macaque physiological data, shape vs. texture cue-conflict settings, images with different visual noises and degradation settings, and the models with/without recurrent visual processing. This reviewer considers that the presented study has great potential to open up the development of more human-like computer vision models based on neural networks. However, there are several points to be addressed and clarified before considering this work to be published.

We are encouraged by Reviewer 2's positive comments and constructive suggestions, and address the specific concerns that were raised below.

Major comments:

1. One of the main questions I had was whether the computer vision model to mimic human vision can be obtained by simply training the model using images with an object with various blur levels. As the authors mentioned multiple times in the manuscript, the human visual acuity systematically declines from the fovea to the periphery. Therefore, this reviewer thinks that the computer vision model should be trained using the images that the human perceives, such as the image with the

high/clear pixel values near the center, and then the pixel values are slowly degraded/blurred from the center to the side.

Can authors show the results from this perspective and compare them with their presented mapping results using the trained CNN model from images with a mixture of clear/blurred images?

As the reviewer suggested, there is potential merit in the notion that computer vision models might derive benefits from incorporating a simulation of foveated vision. One possible implementation could be to present stimuli clearly in the center of an image (akin to a fovea) and to apply progressively stronger levels of blur in the periphery, though it should be noted that this does not entirely match how the human visual system samples the periphery at a much lower spatial resolution. To explore this idea, we trained AlexNet using visual images that were clear at the center (112, 112 pixel location) and increasingly blurred towards the periphery, by linearly increasing the width of a Gaussian filter as a function of eccentricity to a maximum of 8 pixels standard deviation at the outermost radius of a circle in a 224 × 224 image.

We found that exclusive training on these types of images led to inadequate overall top-1 accuracy levels, so we trained AlexNet with a 50/50 mixture of clear images and images with peripheral blur applied. The figure below shows the results of this exploratory analysis, in which peripheral-blur-trained AlexNet's performance was assessed at test without applying peripheral blur. It can be seen that AlexNet trained with peripheral blur leads to better prediction of human fMRI responses in early visual areas (**Figure A, B**), greater bias for shape over texture when tested with clear versions of the cue-conflict images (**E**), and greater robustness to ImageNet-C image degradations on average (**F**). In addition, there appears to be a suggestive trend of better prediction of macaque neuronal responses in areas V2, V4 and IT (**C**).

Supplementary Figure 8. Evaluation of peripheral blur training with AlexNet (yellow). Data obtained from: **A** Xu and Vaziri-Pashkam (2021), **B** Jang et al. (2021), **C** Schrimpf et al. (2020), **D** Cadena et al. (2019), **E** Geirhos et al. (2019), and **F** ImageNet-C, Hendrycks and Dietterich (2019).

These analyses provide further support for our hypothesis; namely that experiences with blurry and low-resolution vision are likely to confer robustness to the visual system. However, we should acknowledge that multiple computational approaches could be adopted to simulate low-resolution peripheral vision and we are not asserting that the above approach is necessarily the most suitable in terms of simulating human vision. That said, the results of this exploratory analysis seem very encouraging and could be of interest to the reader, so we decided to present and discuss these exploratory findings in our revised General Discussion (Lines 458-473 and 543-550 in the manuscript) with the plotted results shown in **Supplementary Figure 8**.

2. Also, ideally, it would be most ecologically valid if the CNN model training and particularly testing could be performed from the image with multiple objects in it while the blur level continuously changes from the center to the side by mimicking the visual scene in the retina. To this end, the multiple objects in the images can first be segmented and then recognized in their categories using the trained CNN models.

We agree that the question of how blur may interact with the observer's point of gaze, especially when one considers the enormous field of view of natural vision, is an interesting one that would be worthwhile to investigate in future studies. Here, we relied on the images and labels readily available in ImageNet to train our CNNs. The majority of these images consist of a single central object with only a single label provided, thus precluding the training approach suggested above. It is conceivable that one might explore this question by training on and evaluating a complex multi-object-labeled dataset, but this would extend beyond the scope of the current project, which already spans 8 CNNs model architectures, multiple training regimes, 4 neuroscience data sets, and several additional test data sets (e.g., ImageNet-C, shape-texture conflict stimuli, Geirhos toolbox).

3. (Paragraph#2, p.4 in Results) “This manipulation was informed by the fact that...”
Related to this point, the corresponding reference seems outdated, so this reviewer is unsure of the validity of the setup of specific target ratios for the frequency of blurry images.

Reference 46 of the original submission consisted of a study by Rovamo et al and is considered both a classic and state-of-the-art measure of how visual sensitivity varies as a function of eccentricity. A series of landmark studies were carried out by Rovamo and Virsu that required several months (or more) of intensive psychophysical testing by the authors who served as the psychophysical observers, and while their results have been replicated in subsequent work, these early pioneering studies are considered the gold standard in the field of human vision science. The review by Strasburger et al. is comprehensive and encompasses many studies. That said, we do acknowledge that mapping between eccentricity and degree of perceived blur relies on simplifying assumptions to make quantitative estimates.

4. (Paragraph#5, p.4 in Results) “Better prediction of cortical responses to high-pass filtered images is notable, as the ...”
Can authors elaborate on this part, as it is not straightforward to understand the reasoning behind it?

We agree that this should be clarified. Reviewer 1 raised a similar question regarding this counterintuitive finding, and an in-depth response can be found where we address the query of Reviewer 1. Here, in brief, we note that early human visual areas (V1-V4) show more confusable fMRI responses to high-pass filtered object images than to the low-pass filtered images, while standard CNNs are excessively sensitive to high spatial frequency information and exhibit very little confusion between the high-pass-filtered images. CNN blur training leads to better alignment to the human cortical responses to high-pass-filtered images. We have added greater discussion of this issue in the revised results (Lines 166-171 in the manuscript) and added **Supplementary Figure 2** to show the

confusion matrices for humans and CNN models.

5. (Paragraph #2, p.13 in Materials and Methods) "..., $P(\cdot)$ is the projection operator onto the ℓ_p ball around x bounded by $\|x - P(x)\|_p \leq \epsilon$, ..."

Can authors elaborate on this method about Projected Gradient Descent-based adversarial examples?

We have included additional information about the methodology in the revised manuscript, with the text indicated below:

(Lines 669 to 684 in Materials and Methods) "A key feature of Projected Gradient Descent is its perturbation limit, which controls the extent of input changes. This constraint is vital for ensuring the practicality of the adversarial examples and for setting a uniform standard for comparison, allowing for the evaluation of diverse models under identical conditions. Specifically, this method generates adversarial examples by iteratively updating gradient-based image perturbations with bounded constraints, as formulated by:

$$x_{t+1} = P(x_t + \alpha \cdot \text{sign}(\nabla_x L(x_t)))$$

where x_t is the perturbed image at t -th step, $P(\cdot)$ is the projection operator to ensure that the adversarial perturbations applied to the image do not exceed a specified threshold, α is the step size, and L is the loss function. The projection operator maps the perturbed image back onto the surface of an ℓ_p -norm ball centered at the original image x and bounded by $\|x - P(x)\|_p \leq \epsilon$. With a random initialization of x , the adversarially perturbed data were generated with 15 iterations using a step size of 0.001. We evaluated both ℓ_∞ and ℓ_2 norm-bounded perturbations with $\epsilon = 0.001$ and 1, respectively."

6. The reasoning behind how the strong-blur CNN can focus on the informative regions of the object identified from the layerwise relevance propagation scheme is not fully straightforward. For example, shape detection can be advantageous when a high-pass filtered image is used compared to the blurred image. This is also perhaps related to the counter-intuitive results on the better prediction for high-pass filtered images using blur-trained CNNs than standard CNNs. Can authors provide their intuition as to why this was the case?

As noted earlier, the reason for the better prediction of neural responses to high-pass filtered images is because human cortical responses to different high-pass filtered objects are quite confusable, and this confusability matrix is better captured by the blur-trained CNNs. Regarding layer-wise relevance propagation analysis, we agree that it is in some sense more descriptive (or depictive) than explanatory. We also acknowledge the potential scenarios where high-pass filtering could be advantageous for identifying an object's shape, especially for highly textured objects. High-pass filtering can also be useful for edge detection.

It can be challenging to figure how a DNN solves a complex objective function (including improved robustness for object recognition), given the large number of non-linear operations that are performed by a DNN. Even with the full series of equations available to be inspected, it can be difficult to ascertain how the successive transformations lead to the final outcome.

That said, we can gain some insight into whether it is the higher or lower spatial frequency components of the cue-conflict stimuli that are responsible for shape vs. texture bias by applying some filtering operations directly to the images themselves; thus, asking if a single linear operation might make a difference or not. As can be seen in the figure below, high-pass filtering does not lead to

a consistent bias effect (top row), some cue-conflict images show an increase in shape bias while others show an increase in textural bias. By contrast, low-pass filtering (with a Gaussian kernel with sigma of 1 or 2 pixels) leads to a highly consistent shift towards shape bias for all cue-conflict images (bottom row). Thus, the increase in shape bias exhibited by our blur-trained CNNs may be explained at least in part to the fact that they tend weight lower spatial frequency information to a greater extent in their classification decisions.

Figure. Assessing the shape bias of standard CNNs using shape-texture conflict stimuli by Geirhos et al. (2019), with input stimuli processed through high-pass filters (**A**) and low-pass filters (**B**). Sigma indicates the pixel value of the standard deviation of the Gaussian used to generate the low-pass filtered images, which in turn could be subtracted away from the clear images to generate the complementary high-pass filtered images.

Minor comments:

7. (Paragraph# 4, p.11 in Materials and Methods) "... and then averaged the results across observers to obtain 8 observations (1 per CNN architecture), ..."

Here, the word, observations must denote the different CNN architecture. If so, it is recommended to use an alternative word since the observer was also used to denote the subject in the same sentence.

Thanks for the suggestion; this part has been edited to remove this ambiguity (Line 595 in the manuscript).

8. Also, the correlational similarity metric used must be spelled out in this sentence.

The term "Pearson correlational similarity" appears just 2 paragraphs above. Also, based on Reviewer 2's question regarding why we use Pearson correlation rather than some other method, such as Spearman correlation, we have added greater discussion of this point to the Methods section (Lines 575-581 in the manuscript), so the correlation method used here should be much clearer now in the revised manuscript.

Reviewer #3 (Remarks to the Author):

This article aims to test the hypothesis that ecological levels of image blur (introduced by retinal sampling and optical defocus) could be instrumental in shaping visual cortex representations. To this end, the authors train a number of convolutional neural networks (CNNs) for image classification with normal, weak-blur or strong-blur image datasets. In support of their hypothesis, they find that blur-trained networks show increased correspondence to human and monkey brain representations in early visual cortex, increased robustness to various types of image perturbations (including adversarial attacks), and increased reliance on shape vs. texture information. These findings are doubly important: they help us understand the role that blur can play in shaping cortical representations; and they help close the gap between machine and biological vision. Therefore, the study should be of interest to both neuroscientists and AI scientists. The paper was very-well written, and the arguments fairly convincing, thus I only have minor suggestions.

We are highly encouraged by Reviewer 3's positive comments, and address the questions raised below.

1. The fact that blur training improves reliance on shape (compared with texture) information was already the point of a recent paper (Yoshihara et al, *Frontiers*, 2023). (I emphasize that the other conclusions on representational similarity and noise/adversarial robustness were not part of the previous study). The authors acknowledge this paper, but describe it as similar to the weak-blur conditions, and suggest that their strong-blur condition yields qualitatively different results. I was not very convinced by this line of reasoning, because the blur training regimes are difficult to compare across the two studies, and the qualitative conclusions (increased reliance on shape, but still short of human behavior) hold for both studies and for both weak-blur and strong-blur training (in other words, the difference is only quantitative). I would much prefer presenting this part of the experiment as a replication of Yoshihara et al's main finding, with an additional conclusion, that shape reliance can be increased by increasing blur (to some extent).

We agree that the training regimes are somewhat different and readily acknowledge this. However, the Yoshihara blur training paradigm involving a combination of clear and blurry images (e.g., B+S-Net, B2S-Net) is more similar to our weak blur condition insofar as the highest blur level used in their study was a sigma of 4 pixels, whereas we used a sigma of up to 5 pixels in our weak blur condition. Yoshihara observed a modest increase in shape bias from a value of ~ 0.4 for a clear-trained version of AlexNet to ~ 0.5 for AlexNet trained on a mixture of clear and blurry images, similar to our weak-blur-trained AlexNet (see new **Supplementary Figure 8**). (We do not consider training with blurry images alone as that leads to a considerable decrease in general classification accuracy.) Admittedly, the frequency of our sigma = 4 and sigma = 5 pixel images was very low in our weak blur condition; that said, we have generally found that the inclusion of a small number of difficult blur images does indeed cause shifts in the representations learned by the CNNs. In comparison, our strong blur condition included images with a sigma of up to 8 pixels, which involves a far more severe level of blur, led to a considerable increase in shape bias in our CNNs (up to 0.72 for strong-blur-trained AlexNet) that exceeded what Yoshihara et al. observed (e.g., B+S-Net with a sigma of 4) as well as our weak-blur CNNs. Regarding how to cite the Yoshihara et al. study, we state in the results section that our weak-blur-trained CNN results "concur with a recent study that reported a similarly modest shift in shape sensitivity after a CNN was trained on a combination of clear and blurry images". We feel this provides appropriate credit to their published work and acknowledges their findings. We also discuss their findings in our general discussion section.

Note that it would be inaccurate for us to claim that our goal was to replicate their work, as we have been working extensively on this project since the summer of 2020, and independently discovered the

impact of mixed blur training on shape bias in the Fall of 2020. The subsequent in-depth neuroscientific studies followed soon after, and the wide-ranging nature of this project and multiple CNNs that were evaluated has taken the well over 2 years to fully assemble.

2. I appreciated the use of a diverse number of CNNs, including Inception and ResNet-based architectures, as well as a recently proposed recurrent model (CORnetS). However, the state-of-the-art in image classification in recent years has shifted from CNNs to Transformer-based architectures. I was curious (and other readers might be as well) about the possibility of extending the present findings to such architectures? Is it just too computationally expensive to retrain a ViT-like model on blurry images? Or are there theoretical reasons why the authors think the approach would not work? If so, this could be addressed in the Discussion.

This is an interesting recommendation made by the reviewer, and we agree that visual transformer models are quickly gaining in popularity in computer vision. Given that the way they process visual information is designed to be more sensitive to spatial relations, it can be interesting to consider whether they too would necessarily benefit from blur training in say, their sensitivity to shape. On the other hand, the vision transformer model, fundamentally rooted in engineering principles, is often regarded as divergent from biological neural networks.

We conducted a preliminary and exploratory analysis of this question using the the Vision Transformer model ViT-B-16 (Dosovitskiy et al., 2020). Below are the results of these analyses. As can be seen, blur-trained ViT provides generally better fits to human fMRI data (**A**, **B**), increased shape bias (**D**) and greater robustness to ImageNet-C. There was no obvious benefit in their ability to account for monkey neurophysiology data, and hints at a possible decrement for the strong-blur-trained ViT model. While not part of our main or original focus, we think that these additional findings may be of interest to readers, and therefore include a brief discussion of these findings near the end of our revised results section (Lines 348-364 in the manuscript). We have also added the figure below as **Supplementary Figure 7** of the manuscript.

Supplementary Figure 7. Evaluation of blur training with visual transformer ViT-B-16. Data obtained from: **A** Xu and Vaziri-Pashkam (2021), **B** Jang et al. (2021), **C** Schrimpf et al. (2020), **D** Geirhos et al. (2019), and **E** ImageNet-C, Hendrycks and Dietterich (2019). The dataset from Cadena et al. (2019) was not included in the evaluation, as it required reducing the input image size to 40×40 pixels, which performs poorly with the ViT architecture.

3. One implication of the findings (targeted to a computer-vision audience) is that it would be a good idea (and a particularly cheap one) to include blurring as a default augmentation strategy for standard image classification libraries. I'm not suggesting that the authors undertake this themselves, but this could be a direct recommendation emanating from the present study?

We appreciate the reviewer's valuable suggestion and feel very encouraged by these comments. We have expanded on our General Discussion section to discuss issues of image augmentation for training CNNs and the potential utility of using blur augmentation as a standard procedure for improving the robustness of CNNs and their consistency with human vision (Changes applied to lines 482-490 in the manuscript).

Reviewer #1 (Remarks to the Author):

I greatly appreciate the revisions made by the authors. The additional analyses and discussions have significantly strengthened their arguments, making the manuscript more persuasive and impactful. The authors have thoroughly addressed the concerns raised during the review. At this point, there's nothing more to add. The paper is well-prepared and is expected to contribute significantly to the field.

Reviewer #1 (Remarks on code availability):

I have briefly reviewed the repository on OSF and noticed that it primarily contains Matlab scripts for plotting figures in the main paper. However, I did not find any raw data files, including the trained weights for the CNN models, nor the codes for training these models. These elements are essential for straightforward replication of the results. While the manuscript does provide sufficient textual details to reproduce the experiments, I am hopeful that the authors will provide these missing components at a later date.

Reviewer #2 (Remarks to the Author):

The authors did an excellent job addressing this reviewer's comments in a concise and very clear manner. I am happy with the revision and strongly suggest the publication in the current form.

Reviewer #3 (Remarks to the Author):

I am fully satisfied with the authors' responses.

REVIEWER COMMENTS

Reviewer #1 (Remarks to the Author):

Key results

This study revealed that training with blurred images helps convolutional neural networks (CNNs) acquire visual representations more similar to biological brains than standard training with only clear images. Moreover, these blur-trained CNNs showed higher robustness to out-of-distribution distortions and higher shape bias than standard CNNs. These results implicate that simulating our experience with blurry retinal images in everyday life is one of key elements for building a computational model of the biological visual system.

Significance

Similar in-silico experiments investigating the effect of training with blurred images have been performed in several previous studies, including those of the authors (all of which are cited). Some of the results reported in this study, such as the improved shape bias and shift of frequency tuning, is not completely new (the authors duly acknowledge this). However, this study stands out from those previous studies in that it examines the effects of blur training from a broader perspective, from comparisons with multiple neurophysiological data to an examination of out-of-distribution robustness, and finds stronger support for the hypothesis that the exposure to blurry images underlies robust biological vision.

Clarity and context

This paper is well motivated and very clearly written. Every paragraph reads well and easy to follow.

We very much appreciate Reviewer 1's comments and encouraging feedback. In writing the manuscript, we did strive to use language to make it accessible to a wider audience.

Reference

I did not find any places where the references were considered inappropriate.

Validity

The authors tested eight different CNN architectures with various depths and with or without recurrence, and found that the results were mostly consistent. Therefore, the results obtained appear to be robust. I found no flaws in the evaluation protocols. The interpretation of the results seems mostly fair (except for the following point).

One thing that I do not understand is why blur training did improve the correlation for high-pass filtered images in the first main result (Fig. 2). Given that the blur-trained CNNs showed tuning shifts in the lower frequency side (Fig. 4), a natural consequence would be to increase the correlation for the low-pass filtered images. Contrary to this expectation, the blur training improved the correlation for the high pass filtered images. The authors only suggested that this could be a sign of out-of-distribution robustness, but this is not convincing enough since the blur-trained CNNs did not show improved correlation for low-pass filtered images.

A possible explanation could be that the early visual areas in the human brain does not represent high-frequency information well enough to discriminate the object categories under test, and that the blur training degraded the high-frequency representation of the CNNs, bringing it somewhat closer to the human representation. This possibility would be testable by computing the representational similarity between the low-pass and high-pass image sets for both human data and CNNs. In addition, it would also be important to see the bandwidth of the frequency tuning of the internal layers of the

CNNs. The current results (Fig. 4) only show the peak frequency, but I wonder if the blur training only shifts the tuning or broadens the bandwidth without sacrificing the high-frequency channels.

We concur with the reviewer's observation that the enhanced correlation observed for fMRI responses to high-pass filtered images deviated from our initial expectations and requires further elucidation. Consistent with the reviewer's insights, we hypothesized that conventional neural networks are excessively biased to process high spatial frequency information for object categorization, a process that diverges from the representational patterns of the human brain.

To investigate this issue in greater detail, we inspected the average representational similarity analysis (RSA) matrices, both for human subjects and CNNs, where we selected the specific CNN layer that most accurately predicted the neural activity of individual human brain regions. As can be seen in the figure below, activity patterns in human visual areas V1-V4 are actually more confusable (i.e., higher off-diagonal correlation values) for high-pass filtered objects than for low-pass filtered objects, whereas standard CNNs exhibit weaker off-diagonal correlations for high-pass filtered objects and much stronger off-diagonal correlations for the low-pass filtered objects. The confusability of the high-pass filtered objects is enhanced in the strong-blur-trained CNNs and the overall RSA pattern of confusability is more similar to that observed in areas V1-V4, which accounts for our observed results. We have added a discussion of the above points to the revised results section (Changes applied to lines 166-171 in the manuscript), and now include **Supplementary Figure 2** of the manuscript to show these RSA results.

Supplementary Figure 2. Plots showing the mean Pearson correlational similarity matrices for human fMRI participants and CNNs in response to the clear, high-pass filtered and low-pass filtered object images used by Xu and Vaziri-Pashkam (2021). The CNN matrices show results for the layer that most accurately predicted neural activity in a specified human visual area.

The reviewer's suggestion to look at possible changes in spatial frequency bandwidth is an interesting one. We performed a follow-up analysis by fitting each individual tuning curve using a Gaussian function on a logarithmic scale and calculating the FWHM of the fitted Gaussian. The bandwidth of the resulting curve was then calculated using the formula $(2 * \text{sqrt}(2 * \log(2)) * \sigma) / \mu$, where μ is the center position of the peak, and σ is the standard deviation of the Gaussian, following the methodology described by David, Hayden, and Gallant (2006). The results indicate a suggestive trend that blur training may lead to broadening of the spatial frequency tuning bandwidth, particularly in the middle and higher layers of the CNNs. We have incorporated these results into **Figure 4B** of the manuscript, where they are presented and discussed alongside with the peak frequency results (Changes applied to lines 205-208, 409-410, and 637-640 in the manuscript).

Figure. Mean bandwidth of the spatial frequency (s.f.) tuning curves for convolutional units from individual layers of standard CNNs (red), weak-blur CNNs (blue), and strong-blur CNNs (purple). Shaded regions indicate 95% confidence intervals.

Data and methodology

All the neurophysiological data (human fMRI responses and monkey neural responses) were taken from the publicly available dataset. The training procedure of CNN models follows standard methodology and is described well in detail to ensure reproducibility.

Analytical approach

I wonder why the authors used Pearson correlation to calculate the similarity between human and CNN RDMs. Pearson correlation assumes a linear relationship between data, but it seems a little too strict to assume such a relationship between CNN and fMRI RDMs. I am curious about whether the similar patterns of results can be obtained when another similarity measure that does not assume linearity, such as Spearman correlation, is used.

We realize that different researchers favor different approaches for comparing and fitting model results to neural data. We have added some text to our revised methods section explaining why we

favor using Pearson correlation for our RSA analyses rather than other approaches such as Spearman correlation (e.g., Kriegeskorte et al., 2008), which does not assume a linear mapping between the predicted and actual responses. That said, we observed very similar results if the same CNN-fMRI analysis is performed using Spearman correlation (see below), and report this fact in the revised methods section (Lines 575-581 in the manuscript).

(Page 13, paragraph #1 in Materials and Methods) "We chose to use Pearson correlation over alternative approaches such as Spearman correlation⁷⁹, as the latter allows for non-linear relationships between predicted and actual response patterns that could allow for excessive model flexibility. Moreover, our analyses of the monkey neurophysiology data relied on linear regression; therefore, the use of Pearson correlation to evaluate the human fMRI data seemed more appropriate. Nevertheless, the pattern of results we observed were almost identical when we applied Spearman correlation instead for our analyses."

Coming from a tradition of early visual processing, our lab favors using a stricter criterion for evaluating the mapping between independent and dependent variables. For example, extensive modeling work has been conducted to describe the contrast-response function observed in cortical areas such as V1; if one were to forego the linear mapping between stimulus contrast and output response, then any model that predicted monotonically increasing activity as a function stimulus strength would be just as good as any other model. We believe that allowing for such non-linear relationships may lead to excess model flexibility, and that in turn could impede one's ability to distinguish between the goodness-of-fit of competing models in certain cases.

Moreover, a substantial portion of our manuscript consists of the evaluation of neuronal data recorded from the macaque monkey. The analytical approach most commonly used in these studies (e.g., BrainScore measures) involves applying a multivariate regression-based approach to map from the CNN model to the observed neuronal responses to each image, assuming a linear mapping. For greater consistency across our analyses of neuronal and fMRI data, we therefore chose to use Pearson correlation for our representational similarity analyses.

Xu and Vaziri-Pashkam (2021)

Jang et al. (2021)

Figure. Spearman correlation-based similarity between CNN model responses and neural responses in different human visual areas to clear, high-pass filtered and low-pass filtered images from Xu and Vaziri-Pashkam (2021) (top) and to clear images, pixelated Gaussian noise, and Fourier phase-scrambled noise from Jang et al. (2021) (bottom). Standard CNNs (red), weak-blur CNNs (blue) and strong-blur CNNs (purple) were evaluated.

I did not find any problems in the other parts of the analysis. However, as I have no experience in fMRI data analysis by myself, I would like to leave the rigorous review of that part to another reviewer.

Suggested improvements

The paper focuses on the similarity of the internal representations of CNNs and brains, but it is also interesting to see how close the blur trained CNNs are to humans at the behavior level. The current results suggested that the stronger the blur size of images experienced during training, the more similar the representation of the CNNs is to the brain. Could something similar be observed at the level of object recognition performance?

A similar question arises in the results of the out-of-distribution robustness test using ImageNet-C. Here, the authors demonstrated that the blur-trained CNNs exhibit better classification performance than the standard-trained ones for several out-of-distribution distortion types, but without discussing the extent to which humans can classify object categories under these distortions.

As a way to address the above questions, I would suggest using a publicly available toolbox developed to benchmark the similarity between humans and machine learning models:

<https://github.com/bethgelab/model-vs-human>

Reference: Geirhos et al. Partial success in closing the gap between human and machine vision. NeurIPS2021.

The beauty of this toolbox is that it can evaluate image-level error consistency, a finer measure than classification accuracy.

We thank the reviewer for this very helpful suggestion. Based on this recommendation, we downloaded the toolbox and evaluated our full set of models, which were trained from scratch using 1000-category ImageNet training with standard, weak blur, or strong blur conditions. As can be seen in the figure below, our blur-trained CNNs consistently outperform CNNs trained using standard clear images with respect to classification accuracy (i.e., labeled out-of-distribution accuracy; note that while low-pass filtered images as a test set are not strictly out-of-distribution, we retain the terminology from Geirhos et al., 2021). Moreover, they show improved consistency and improved error consistency when compared to the human performance data collected by Geirhos et al. (2021). We have added a new section to our results (see *Correspondence with human behavioral responses to out-of-distribution data*; lines 308-322 and 686-702 in the manuscript) that discusses our evaluation of the models using the toolbox. In this section, we introduce a new figure (**Figure 8**) in the revised manuscript, as further evidence that blur training does indeed improve the correspondence between deep neural network models and human vision.

Figure 8. A Classification accuracy for standard (red), weak-blur (blue) and strong-blur CNNs (purple) based on aggregated performance for 17 out-of-distribution datasets provided by Geirhos et al. (2021). Note that 1 of 17 conditions involved blurry images, which was not out-of-distribution for the blur-trained CNNs. **B** Accuracy difference between humans and CNN models. **C, D** Consistency of responses and error responses between humans and CNNs; higher values indicate better human-AI alignment, with gray bars indicating human-to-human consistency.

Another point that remains unclear is how special blur distortion is compared to other data augmentations such as Gaussian noise and adversarial noise, both of which are known to produce low-frequency bias (Yin et al., A Fourier Perspective on Model Robustness in Computer Vision, NeurIPS2019). While I understand that the blur-training is more biologically plausible, it would be beneficial for the machine learning community to compare and quantify the effectiveness of blur training with the other types of data augmentation. Although the authors repeatedly emphasize the superiority of the blur training over others that it can lead to better generalization to out-of-distribution performance, it is difficult to prove this without comparing models trained under the same procedure, varying only the augmentation type and strength. In fact, a previous study (Geirhos et al., 2018, Generalisation in humans and deep neural networks), which the authors cited to show the weak generalization performance of other augmentations, also used low-pass filtering for training, but did not observe the similar amount of generalization as presented in the current study. It would be nice if the authors could clarify what is responsible for this difference.

The reviewer raises a number of interesting points about other options for data augmentation and whether training with Gaussian or adversarial noise might lead to similar improvements in generalization performance as blur training. Our original submission included some discussion of these points, and in our revision, we have expanded our discussion of these issues and related studies. In our recently published work in PLOS Biology (Jang et al., 2021), we found that CNNs trained to recognize objects in Gaussian noise also exhibit successful generalization to other forms of visual noise (e.g., salt and pepper noise) as well as better recognition of low-pass filtered images, in

general agreement with the above-mentioned study by Yin et al. but apparently in contrast to the report by Geirhos et al. While humans certainly do encounter noise elements on occasion (e.g., snow, dust, rain), they encounter visual blur far more often, and for these reasons, we posit that it is more likely that real-world experiences with blur confer some robustness to noise rather than the other way around. However, we acknowledge that it would be very useful for future studies to obtain estimates of the frequency of blur, noise and other forms of image degradation in everyday human vision and characterize the visual statistics of those conditions. We have revised our discussion section to address these issues to a greater extent (Changes applied to lines 396, 451-456 in the manuscript).

The fact that our blur-trained CNNs exhibit successful generalization to out-of-distribution conditions, including the behavioral conditions that Geirhos et al (2021) evaluated, further demonstrates that our blur-trained networks do well at generalizing to other challenging visual conditions such as those involving visual noise. We are highly confident of our model results as we evaluated multiple CNN models in this study and observed consistent improvement in generalization performance for all blur-trained models, including ResNet50 which was the primary model evaluated by Geirhos et al. (2018). Note that multiple aspects of our training protocol differed from Geirhos et al., and it is conceivable that any of these differences (or others we might not be aware of) could account for the lack of generalization that Geirhos et al. (2018) observed. These include the fact that we used separate object images for training and test while Geirhos et al. used the same object images at test (after applying image degradation); also, we trained CNNs with all 1000 object categories in ImageNet while Geirhos et al. used a more limited set of 16 object categories.

While we agree that it would be of interest for future studies to investigate why Geirhos et al. found weak generalization to out-of-distribution conditions, this question falls outside of the main goals of our study, which is to report how blur-training leads to broad-ranging improvements in CNN performance and better alignment with the human visual system. Also, we would be hesitant to directly speculate in our manuscript as to why Geirhos et al. observed effects of DNN training that appear consistent with potential problems of overfitting.

Reviewer #2 (Remarks to the Author):

This manuscript presented very interesting work on the improvement of convolutional neural networks in order to match the neuronal processes of human vision. The strength of the work is that the authors showed the validity of their proposed models from a wide range of systematical evaluations of various datasets and settings from the human fMRI to macaque physiological data, shape vs. texture cue-conflict settings, images with different visual noises and degradation settings, and the models with/without recurrent visual processing. This reviewer considers that the presented study has great potential to open up the development of more human-like computer vision models based on neural networks. However, there are several points to be addressed and clarified before considering this work to be published.

We are encouraged by Reviewer 2's positive comments and constructive suggestions, and address the specific concerns that were raised below.

Major comments:

1. One of the main questions I had was whether the computer vision model to mimic human vision can be obtained by simply training the model using images with an object with various blur levels. As the authors mentioned multiple times in the manuscript, the human visual acuity systematically declines from the fovea to the periphery. Therefore, this reviewer thinks that the computer vision model should be trained using the images that the human perceives, such as the image with the

high/clear pixel values near the center, and then the pixel values are slowly degraded/blurred from the center to the side.

Can authors show the results from this perspective and compare them with their presented mapping results using the trained CNN model from images with a mixture of clear/blurred images?

As the reviewer suggested, there is potential merit in the notion that computer vision models might derive benefits from incorporating a simulation of foveated vision. One possible implementation could be to present stimuli clearly in the center of an image (akin to a fovea) and to apply progressively stronger levels of blur in the periphery, though it should be noted that this does not entirely match how the human visual system samples the periphery at a much lower spatial resolution. To explore this idea, we trained AlexNet using visual images that were clear at the center (112, 112 pixel location) and increasingly blurred towards the periphery, by linearly increasing the width of a Gaussian filter as a function of eccentricity to a maximum of 8 pixels standard deviation at the outermost radius of a circle in a 224×224 image.

We found that exclusive training on these types of images led to inadequate overall top-1 accuracy levels, so we trained AlexNet with a 50/50 mixture of clear images and images with peripheral blur applied. The figure below shows the results of this exploratory analysis, in which peripheral-blur-trained AlexNet's performance was assessed at test without applying peripheral blur. It can be seen that AlexNet trained with peripheral blur leads to better prediction of human fMRI responses in early visual areas (**Figure A, B**), greater bias for shape over texture when tested with clear versions of the cue-conflict images (**E**), and greater robustness to ImageNet-C image degradations on average (**F**). In addition, there appears to be a suggestive trend of better prediction of macaque neuronal responses in areas V2, V4 and IT (**C**).

Supplementary Figure 8. Evaluation of peripheral blur training with AlexNet (yellow). Data obtained from: **A** Xu and Vaziri-Pashkam (2021), **B** Jang et al. (2021), **C** Schrimpf et al. (2020), **D** Cadena et al. (2019), **E** Geirhos et al. (2019), and **F** ImageNet-C, Hendrycks and Dietterich (2019).

These analyses provide further support for our hypothesis; namely that experiences with blurry and low-resolution vision are likely to confer robustness to the visual system. However, we should acknowledge that multiple computational approaches could be adopted to simulate low-resolution peripheral vision and we are not asserting that the above approach is necessarily the most suitable in terms of simulating human vision. That said, the results of this exploratory analysis seem very encouraging and could be of interest to the reader, so we decided to present and discuss these exploratory findings in our revised General Discussion (Lines 458-473 and 543-550 in the manuscript) with the plotted results shown in **Supplementary Figure 8**.

2. Also, ideally, it would be most ecologically valid if the CNN model training and particularly testing could be performed from the image with multiple objects in it while the blur level continuously changes from the center to the side by mimicking the visual scene in the retina. To this end, the multiple objects in the images can first be segmented and then recognized in their categories using the trained CNN models.

We agree that the question of how blur may interact with the observer's point of gaze, especially when one considers the enormous field of view of natural vision, is an interesting one that would be worthwhile to investigate in future studies. Here, we relied on the images and labels readily available in ImageNet to train our CNNs. The majority of these images consist of a single central object with only a single label provided, thus precluding the training approach suggested above. It is conceivable that one might explore this question by training on and evaluating a complex multi-object-labeled dataset, but this would extend beyond the scope of the current project, which already spans 8 CNNs model architectures, multiple training regimes, 4 neuroscience data sets, and several additional test data sets (e.g., ImageNet-C, shape-texture conflict stimuli, Geirhos toolbox).

3. (Paragraph#2, p.4 in Results) “This manipulation was informed by the fact that...”
Related to this point, the corresponding reference seems outdated, so this reviewer is unsure of the validity of the setup of specific target ratios for the frequency of blurry images.

Reference 46 of the original submission consisted of a study by Rovamo et al and is considered both a classic and state-of-the-art measure of how visual sensitivity varies as a function of eccentricity. A series of landmark studies were carried out by Rovamo and Virsu that required several months (or more) of intensive psychophysical testing by the authors who served as the psychophysical observers, and while their results have been replicated in subsequent work, these early pioneering studies are considered the gold standard in the field of human vision science. The review by Strasburger et al. is comprehensive and encompasses many studies. That said, we do acknowledge that mapping between eccentricity and degree of perceived blur relies on simplifying assumptions to make quantitative estimates.

4. (Paragraph#5, p.4 in Results) “Better prediction of cortical responses to high-pass filtered images is notable, as the ...”
Can authors elaborate on this part, as it is not straightforward to understand the reasoning behind it?

We agree that this should be clarified. Reviewer 1 raised a similar question regarding this counterintuitive finding, and an in-depth response can be found where we address the query of Reviewer 1. Here, in brief, we note that early human visual areas (V1-V4) show more confusable fMRI responses to high-pass filtered object images than to the low-pass filtered images, while standard CNNs are excessively sensitive to high spatial frequency information and exhibit very little confusion between the high-pass-filtered images. CNN blur training leads to better alignment to the human cortical responses to high-pass-filtered images. We have added greater discussion of this issue in the revised results (Lines 166-171 in the manuscript) and added **Supplementary Figure 2** to show the

confusion matrices for humans and CNN models.

5. (Paragraph #2, p.13 in Materials and Methods) "..., $P(\cdot)$ is the projection operator onto the ℓ_p ball around x bounded by $\|x - P(x)\|_p \leq \epsilon$, ..."

Can authors elaborate on this method about Projected Gradient Descent-based adversarial examples?

We have included additional information about the methodology in the revised manuscript, with the text indicated below:

(Lines 669 to 684 in Materials and Methods) "A key feature of Projected Gradient Descent is its perturbation limit, which controls the extent of input changes. This constraint is vital for ensuring the practicality of the adversarial examples and for setting a uniform standard for comparison, allowing for the evaluation of diverse models under identical conditions. Specifically, this method generates adversarial examples by iteratively updating gradient-based image perturbations with bounded constraints, as formulated by:

$$x_{t+1} = P(x_t + \alpha \cdot \text{sign}(\nabla_x L(x_t)))$$

where x_t is the perturbed image at t -th step, $P(\cdot)$ is the projection operator to ensure that the adversarial perturbations applied to the image do not exceed a specified threshold, α is the step size, and L is the loss function. The projection operator maps the perturbed image back onto the surface of an ℓ_p -norm ball centered at the original image x and bounded by $\|x - P(x)\|_p \leq \epsilon$. With a random initialization of x , the adversarially perturbed data were generated with 15 iterations using a step size of 0.001. We evaluated both ℓ_∞ and ℓ_2 norm-bounded perturbations with $\epsilon = 0.001$ and 1, respectively."

6. The reasoning behind how the strong-blur CNN can focus on the informative regions of the object identified from the layerwise relevance propagation scheme is not fully straightforward. For example, shape detection can be advantageous when a high-pass filtered image is used compared to the blurred image. This is also perhaps related to the counter-intuitive results on the better prediction for high-pass filtered images using blur-trained CNNs than standard CNNs. Can authors provide their intuition as to why this was the case?

As noted earlier, the reason for the better prediction of neural responses to high-pass filtered images is because human cortical responses to different high-pass filtered objects are quite confusable, and this confusability matrix is better captured by the blur-trained CNNs. Regarding layer-wise relevance propagation analysis, we agree that it is in some sense more descriptive (or depictive) than explanatory. We also acknowledge the potential scenarios where high-pass filtering could be advantageous for identifying an object's shape, especially for highly textured objects. High-pass filtering can also be useful for edge detection.

It can be challenging to figure how a DNN solves a complex objective function (including improved robustness for object recognition), given the large number of non-linear operations that are performed by a DNN. Even with the full series of equations available to be inspected, it can be difficult to ascertain how the successive transformations lead to the final outcome.

That said, we can gain some insight into whether it is the higher or lower spatial frequency components of the cue-conflict stimuli that are responsible for shape vs. texture bias by applying some filtering operations directly to the images themselves; thus, asking if a single linear operation might make a difference or not. As can be seen in the figure below, high-pass filtering does not lead to

a consistent bias effect (top row), some cue-conflict images show an increase in shape bias while others show an increase in textural bias. By contrast, low-pass filtering (with a Gaussian kernel with sigma of 1 or 2 pixels) leads to a highly consistent shift towards shape bias for all cue-conflict images (bottom row). Thus, the increase in shape bias exhibited by our blur-trained CNNs may be explained at least in part to the fact that they tend weight lower spatial frequency information to a greater extent in their classification decisions.

Figure. Assessing the shape bias of standard CNNs using shape-texture conflict stimuli by Geirhos et al. (2019), with input stimuli processed through high-pass filters (**A**) and low-pass filters (**B**). Sigma indicates the pixel value of the standard deviation of the Gaussian used to generate the low-pass filtered images, which in turn could be subtracted away from the clear images to generate the complementary high-pass filtered images.

Minor comments:

7. (Paragraph# 4, p.11 in Materials and Methods) "... and then averaged the results across observers to obtain 8 observations (1 per CNN architecture), ..."

Here, the word, observations must denote the different CNN architecture. If so, it is recommended to use an alternative word since the observer was also used to denote the subject in the same sentence.

Thanks for the suggestion; this part has been edited to remove this ambiguity (Line 595 in the manuscript).

8. Also, the correlational similarity metric used must be spelled out in this sentence.

The term "Pearson correlational similarity" appears just 2 paragraphs above. Also, based on Reviewer 2's question regarding why we use Pearson correlation rather than some other method, such as Spearman correlation, we have added greater discussion of this point to the Methods section (Lines 575-581 in the manuscript), so the correlation method used here should be much clearer now in the revised manuscript.

Reviewer #3 (Remarks to the Author):

This article aims to test the hypothesis that ecological levels of image blur (introduced by retinal sampling and optical defocus) could be instrumental in shaping visual cortex representations. To this end, the authors train a number of convolutional neural networks (CNNs) for image classification with normal, weak-blur or strong-blur image datasets. In support of their hypothesis, they find that blur-trained networks show increased correspondence to human and monkey brain representations in early visual cortex, increased robustness to various types of image perturbations (including adversarial attacks), and increased reliance on shape vs. texture information. These findings are doubly important: they help us understand the role that blur can play in shaping cortical representations; and they help close the gap between machine and biological vision. Therefore, the study should be of interest to both neuroscientists and AI scientists. The paper was very-well written, and the arguments fairly convincing, thus I only have minor suggestions.

We are highly encouraged by Reviewer 3's positive comments, and address the questions raised below.

1. The fact that blur training improves reliance on shape (compared with texture) information was already the point of a recent paper (Yoshihara et al, *Frontiers*, 2023). (I emphasize that the other conclusions on representational similarity and noise/adversarial robustness were not part of the previous study). The authors acknowledge this paper, but describe it as similar to the weak-blur conditions, and suggest that their strong-blur condition yields qualitatively different results. I was not very convinced by this line of reasoning, because the blur training regimes are difficult to compare across the two studies, and the qualitative conclusions (increased reliance on shape, but still short of human behavior) hold for both studies and for both weak-blur and strong-blur training (in other words, the difference is only quantitative). I would much prefer presenting this part of the experiment as a replication of Yoshihara et al's main finding, with an additional conclusion, that shape reliance can be increased by increasing blur (to some extent).

We agree that the training regimes are somewhat different and readily acknowledge this. However, the Yoshihara blur training paradigm involving a combination of clear and blurry images (e.g., B+S-Net, B2S-Net) is more similar to our weak blur condition insofar as the highest blur level used in their study was a sigma of 4 pixels, whereas we used a sigma of up to 5 pixels in our weak blur condition. Yoshihara observed a modest increase in shape bias from a value of ~ 0.4 for a clear-trained version of AlexNet to ~ 0.5 for AlexNet trained on a mixture of clear and blurry images, similar to our weak-blur-trained AlexNet (see new **Supplementary Figure 8**). (We do not consider training with blurry images alone as that leads to a considerable decrease in general classification accuracy.) Admittedly, the frequency of our sigma = 4 and sigma = 5 pixel images was very low in our weak blur condition; that said, we have generally found that the inclusion of a small number of difficult blur images does indeed cause shifts in the representations learned by the CNNs. In comparison, our strong blur condition included images with a sigma of up to 8 pixels, which involves a far more severe level of blur, led to a considerable increase in shape bias in our CNNs (up to 0.72 for strong-blur-trained AlexNet) that exceeded what Yoshihara et al. observed (e.g., B+S-Net with a sigma of 4) as well as our weak-blur CNNs. Regarding how to cite the Yoshihara et al. study, we state in the results section that our weak-blur-trained CNN results "concur with a recent study that reported a similarly modest shift in shape sensitivity after a CNN was trained on a combination of clear and blurry images". We feel this provides appropriate credit to their published work and acknowledges their findings. We also discuss their findings in our general discussion section.

Note that it would be inaccurate for us to claim that our goal was to replicate their work, as we have been working extensively on this project since the summer of 2020, and independently discovered the

impact of mixed blur training on shape bias in the Fall of 2020. The subsequent in-depth neuroscientific studies followed soon after, and the wide-ranging nature of this project and multiple CNNs that were evaluated has taken the well over 2 years to fully assemble.

2. I appreciated the use of a diverse number of CNNs, including Inception and ResNet-based architectures, as well as a recently proposed recurrent model (CORnetS). However, the state-of-the-art in image classification in recent years has shifted from CNNs to Transformer-based architectures. I was curious (and other readers might be as well) about the possibility of extending the present findings to such architectures? Is it just too computationally expensive to retrain a ViT-like model on blurry images? Or are there theoretical reasons why the authors think the approach would not work? If so, this could be addressed in the Discussion.

This is an interesting recommendation made by the reviewer, and we agree that visual transformer models are quickly gaining in popularity in computer vision. Given that the way they process visual information is designed to be more sensitive to spatial relations, it can be interesting to consider whether they too would necessarily benefit from blur training in say, their sensitivity to shape. On the other hand, the vision transformer model, fundamentally rooted in engineering principles, is often regarded as divergent from biological neural networks.

We conducted a preliminary and exploratory analysis of this question using the the Vision Transformer model ViT-B-16 (Dosovitskiy et al., 2020). Below are the results of these analyses. As can be seen, blur-trained ViT provides generally better fits to human fMRI data (**A**, **B**), increased shape bias (**D**) and greater robustness to ImageNet-C. There was no obvious benefit in their ability to account for monkey neurophysiology data, and hints at a possible decrement for the strong-blur-trained ViT model. While not part of our main or original focus, we think that these additional findings may be of interest to readers, and therefore include a brief discussion of these findings near the end of our revised results section (Lines 348-364 in the manuscript). We have also added the figure below as **Supplementary Figure 7** of the manuscript.

Supplementary Figure 7. Evaluation of blur training with visual transformer ViT-B-16. Data obtained from: **A** Xu and Vaziri-Pashkam (2021), **B** Jang et al. (2021), **C** Schrimpf et al. (2020), **D** Geirhos et al. (2019), and **E** ImageNet-C, Hendrycks and Dietterich (2019). The dataset from Cadena et al. (2019) was not included in the evaluation, as it required reducing the input image size to 40×40 pixels, which performs poorly with the ViT architecture.

3. One implication of the findings (targeted to a computer-vision audience) is that it would be a good idea (and a particularly cheap one) to include blurring as a default augmentation strategy for standard image classification libraries. I'm not suggesting that the authors undertake this themselves, but this could be a direct recommendation emanating from the present study?

We appreciate the reviewer's valuable suggestion and feel very encouraged by these comments. We have expanded on our General Discussion section to discuss issues of image augmentation for training CNNs and the potential utility of using blur augmentation as a standard procedure for improving the robustness of CNNs and their consistency with human vision (Changes applied to lines 482-490 in the manuscript).